# H3K9me3-mediated epigenetic regulation of senescence in mice predicts outcome of lymphoma patients

Kolja Schleich[1,10], Julia Kase[1,10], Jan R. Dörr[1,10], Saskia Trescher[2], Animesh Bhattacharya[1], Yong Yu[3], Elizabeth M. Wailes[1], Dorothy N. Y. Fan[1,4], Philipp Lohneis[5], Maja Milanovic [1], Andrea Lau[1], Dido Lenze[6], Michael Hummel[4,6], Bjoern Chapuy[7], Ulf Leser[2], Maurice Reimann[1], Soyoung Lee [1,3,4] & Clemens A. Schmitt [1,3,4,8,9✉]

Lesion-based targeting strategies underlie cancer precision medicine. However, biological principles – such as cellular senescence – remain difficult to implement in molecularly informed treatment decisions. Functional analyses in syngeneic mouse models and cross-species validation in patient datasets might uncover clinically relevant genetics of biological response programs. Here, we show that chemotherapy-exposed primary Eμ-*myc* transgenic lymphomas – with and without defined genetic lesions – recapitulate molecular signatures of patients with diffuse large B-cell lymphoma (DLBCL). Importantly, we interrogate the murine lymphoma capacity to senesce and its epigenetic control via the histone H3 lysine 9 (H3K9)-methyltransferase Suv(ar)39h1 and H3K9me3-active demethylases by loss- and gain-of-function genetics, and an unbiased clinical trial-like approach. A mouse-derived senescence-indicating gene signature, termed "SUVARness", as well as high-level H3K9me3 lymphoma expression, predict favorable DLBCL patient outcome. Our data support the use of functional genetics in transgenic mouse models to incorporate basic biology knowledge into cancer precision medicine in the clinic.

[1] Charité – University Medical Center, Department of Hematology, Oncology and Tumor Immunology, Virchow Campus, and Molekulares Krebsforschungszentrum, Augustenburger Platz 1, 13353 Berlin, Germany. [2] Institute for Computer Science, Humboldt-Universität zu Berlin, Unter Den Linden 6, 10099 Berlin, Germany. [3] Max-Delbrück-Center for Molecular Medicine in the Helmholtz Association, Robert-Rössle-Straße 10, 13125 Berlin, Germany. [4] Deutsches Konsortium für Translationale Krebsforschung (German Cancer Consortium), Partner Site Berlin, Berlin, Germany. [5] University Hospital Cologne, Pathology, Kerpener Straße 62, 50937 Cologne, Germany. [6] Charité – University Medical Center, Pathology, Charitéplatz 1, 10117 Berlin, Germany. [7] University Medical Center Göttingen, Department of Hematology and Medical Oncology, Robert-Koch-Straße 40, 37075 Göttingen, Germany. [8] Kepler University Hospital, Department of Hematology and Oncology, Johannes Kepler University, Krankenhausstraße 9, 4020 Linz, Austria. [9] Berlin Institute of Health, Anna-Louisa-Karsch-Straße 2, 10178 Berlin, Germany. [10] These authors contributed equally: Kolja Schleich, Julia Kase, Jan R. Dörr. ✉email: clemens. schmitt@charite.de

Advanced stage solid tumors that are no longer solely surgically curable, and most hematological malignancies, in principle, require systemic pharmacological anti-cancer therapies[1]. Irrespective of their disseminated growth, some leukemia and lymphoma entities are potentially curable by chemotherapy, with treatment resistance, in turn, representing the key determinant of patient death in hematological malignancies. Although mostly developed on an empirical basis over many decades, DNA-damaging agents have significant efficacy and will remain the backbone of anti-cancer therapies for many tumor entities in the future, and this especially applies to aggressive lymphoma such as diffuse large B-cell lymphoma (DLBCL). While standard "R-CHOP" immune-chemotherapy (i.e., the anti-CD20 antibody rituximab plus cyclophosphamide [CTX], adriamycin [ADR], vincristine, and prednisone[2]) achieves long-term disease control in about 60% of the DLBCL patients[3,4], the outcome of failing patients is, despite intense salvage regimens, dismal[5]. Since the beginning of the millennium, transcriptome- and genome-based profiling of DLBCL has led to much deeper molecular insights in this entity[6–15], but virtually all recent randomized phase III trials, intended to enhance R-CHOP efficacy by an additional agent and conducted in largely unselected DLBCL patient populations, failed. Hence, the functional impact of distinct mutations, molecular subtypes, and, in particular, global therapy effector programs—namely apoptosis and senescence—on patient long-term outcome remains, in general, rather poorly understood and therapeutically underexploited.

Predicting drug responses and deciphering treatment failure is of pivotal importance to improve treatment outcome in oncology. Preclinical platforms for testing anti-cancer drug susceptibility include cytotoxicity assays in established multi-passage cancer cell lines, primary tumor cells grown in 2D- or 3D-cultures, patient-derived xenograft (PDX) models in immunocompromised mice, or syngeneic, orthotopically transplantable tumor models in mice with normal immune functions[16–19]. We investigated here the Eμ-myc transgenic mouse as a tractable and immune-competent model approximation to DLBCL that allowed us to study treatment responses in a larger number of primary lymphomas in a clinical trial-like fashion. Specifically, Eμ-myc transgenic lymphomas exhibit clinical, histopathological and genetic features of aggressive human B-cell lymphoma, are transplantable, and grow system-wide at natural sites—i.e., typically in the lymph nodes (LN), the spleen, the BM, and the peripheral blood, sometimes also in visceral organs—where they shape their microenvironments and engage in host immune interactions[20–23]. Upon lymphoma manifestation and administration of therapy, drug, or radiation responses can be easily monitored by clinical palpation of LN sites or whole-body imaging techniques, especially if lymphoma cells (LC) are stably fluorescence- or bioluminescence-engineered[24,25]. In the past, we utilized this model to investigate the impact of candidate genes or global effector programs, with particular interest in cellular senescence, on treatment outcome via reverse genetics[20–22,25–28].

Stress response programs such as apoptosis and cellular senescence serve as important effector principles of anti-cancer therapy[29]. Cellular senescence is an acutely stress-inducible cell-cycle arrest condition that complements apoptosis as another ultimate cell-cycle exit program[30–32]. The firm G1-phase arrest of oncogene- or therapy-induced senescent (OIS, TIS, respectively) cells is executed through the Retinoblastoma protein (Rb)/E2F transcription factor-guided trimethylation at the lysine-9 residue of histone H3 (H3K9me3), mediated, for instance, by the indirectly Rb-bound H3K9 methyltransferase Suv39h1 (suppressor of variegation 3–9 homolog 1, briefly "suvar"), thereby creating a repressive chromatin environment in the vicinity of S-phase-promoting E2F target genes[33,34]. However, dissecting TIS as a key

contributor to long-term outcome, and, even more, anticipating the senescence response to a future therapy are particularly difficult in primary patient material, underscoring the need for functional investigations in patient-predictive mouse models of cancer[35,36].

Hence, we explore here a transgenic mouse lymphoma model in an unbiased forward genetics approach, dissect and compare underlying genetics of drug (in)sensitivity across species, evaluate cellular senescence as a drug effector program in the mouse, and probe mouse model-derived senescence mediators and related genetic classifiers as biomarkers of DLBCL patient survival—functional investigations technically and ethically virtually impossible to be solely conducted in cancer patients. In essence, our approach seeks to bioinformatically extract predictive signatures from outcome analyses in murine aggressive B-cell lymphoma models to inform precision medicine in DLBCL patients.

## Results

**Eμ-myc lymphomas recapitulate DLBCL treatment outcome.** To obtain array-based genome-wide gene expression profiles (GEP) and document treatment responses of individual lymphomas, a larger series ($n = 39$) of primary B-cell lymphomas that arose in Eμ-myc transgenic mice (hereafter referred to as "control" lymphomas) were intravenously transplanted into two wild-type recipient mice each (Fig. 1a). At the time well-palpable LN enlargements had formed, a single dose of the alkylating chemo-agent CTX was intraperitoneally administered, and responses were monitored at least twice a week. Consistent with the high initial chemo-sensitivity of most lymphoma patients to their first-line induction therapy, all mice achieved a complete remission (CR) by clinical criteria (see Methods for details) within a few days after CTX application, collectively forming the treatment-sensitive ["SENS"] group of primary lymphomas at diagnosis (Fig. 1a). In the range of the progressing fraction of DLBCL patients treated with standard R-CHOP induction therapy, 35 of the 78 mice presented with re-growing LN during the 100-day observation period (i.e. the "relapse-prone [RP]" group), while about half of the cohort (i.e., the NR group) remained relapse-free (comparable to the 47% 5-year event-free survival for R-CHOP patients in major clinical trials[37,38]), and was considered cured, therefore designated "never relapse [NR]" lymphomas (Fig. 1b, plateau of the green curve). Interestingly, time-to-relapse (TTR) was very similar among most of those paired recipients that were transplanted with the same primary lymphoma (Supplementary Fig. 1a), indicating that the key information determining treatment outcome is encoded in the tumor cell population, albeit not excluding an instructive impact the tumor cells may have on other host components. When retreated at the time of relapse with a second application of CTX, all mice entered a CR again, but none achieved long-term disease control anymore (Fig. 1b, blue curve). Moreover, the recapitulation of the standard CHOP regimen in our lymphoma-bearing mouse cohort produced the same long-term outcome: virtually every primary lymphoma that relapsed after CTX also recurred after CHOP, and lymphomas achieving lasting remissions in response to CHOP were also cured by CTX alone (Supplementary Fig. 1b). Mice reprogressing after two rounds of CTX were subjected to a third dose of the same agent, again resulting in even shorter durations of response and no lasting disease control (Fig. 1b, red curve). Eventually, if exposed to additional rounds of CTX, mice exhibited at best a "no change" situation or presented with "progressive disease", i.e., no shrinkage or even continuous growth despite therapy. In essence, we established a transgenic lymphoma treatment platform that resembles some clinical results observed in DLBCL patient cohorts in response to CHOP-based therapies,

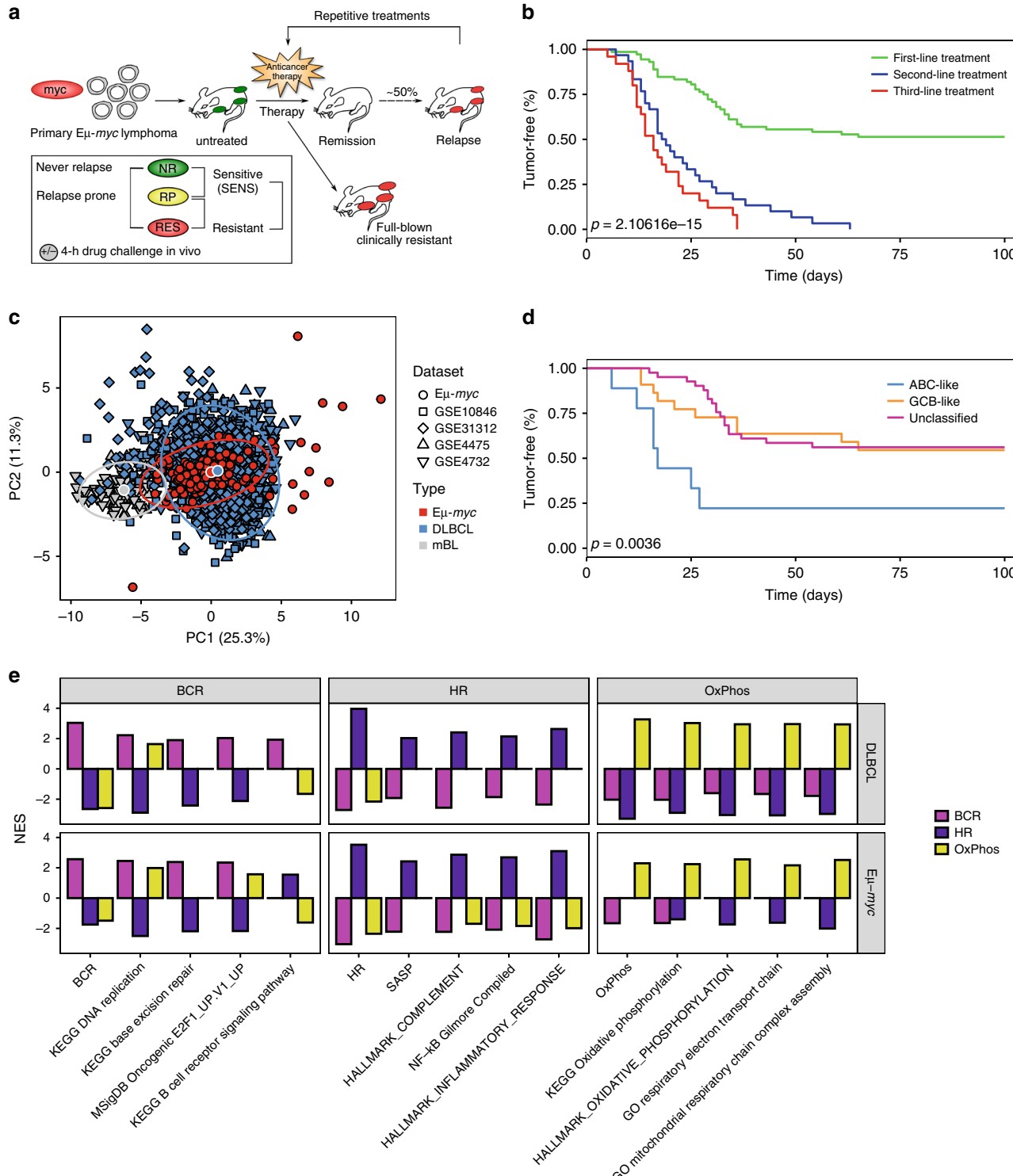

**Fig. 1 The Eµ-*myc* mouse lymphoma model recapitulates molecular signatures and some clinical features of human DLBCL. a** Utilization of the Eµ-*myc* model in a clinical trial-like fashion. Eµ-*myc* lymphomas were transplanted into wild-type recipients and treated with CTX upon lymphoma formation. After initial remission, tumors that relapsed (RP, unlike those that did not [NR]) were retreated. Lymphomas were considered clinically resistant after a third relapse (RES). **b** Tumor-free survival of CTX-treated mice after first(-line) treatment (*n* = 76, green line), after second treatment (*n* = 31, blue line), and after third treatment (*n* = 28, red line). Two mice per lymphoma were monitored for response. **c** Principal components analysis based on signature genes separating patients with Burkitt's lymphoma (*n* = 98) from those with DLBCL (*n* = 1305) in different datasets and Eµ-*myc* lymphomas (*n* = 154). Each dataset was separately row (gene) mean-centered and scaled to unit variance before being combined in a single dataset. The larger white-bordered points denote the centroid of each lymphoma type. **d** Tumor-free survival of CTX-treated mice bearing ABC-like (*n* = 10), GCB-like (*n* = 24) or unclassified (*n* = 42) Eµ-*myc* lymphomas. Two mice per lymphoma were used. **e** Pattern comparison of multiple GSEA results of the three distinct CCC subgroups (as indicated) with one compared against the two others for DLBCL cases and Eµ-*myc* lymphomas. Each column shows group-specific upregulated gene sets. The color code denotes analyses of each CCC group against the rest. NES normalized enrichment score.

and that culminates, if no cure is achieved, in a progressive shortening of response durations, ultimately leading to full-blown chemoresistance ("RES"; Fig. 1a).

**DLBCL classifiers delineate corresponding murine subgroups.** Transcriptome analyses unveiled two distinct cell-of-origin (COO) signatures in DLBCL, representing a germinal center B-cell (GCB) and an activated B-cell (ABC) subtype[6,8], with an inferior outcome consistently reported for the ABC subtype in many trials applying CHOP or R-CHOP regimens to DLBCL patients[8]. We previously recapitulated COO-related differences in NF-κB activity in Eμ-myc lymphomas[27,39]. To enhance statistical robustness and to increase biological diversity, we combined GEP data from our 39 primary lymphomas with a published GEP set obtained from 115 additional primary Eμ-myc lymphomas[40], comprising a total of 154 samples. Using principal component analysis and the murine homologue of a human gene classifier built to distinguish DLBCL from Burkitt's lymphoma (BL)[41], we first verified that Eμ-myc lymphomas share transcriptional features with DLBCL and are distinct from molecular BL (Fig. 1c). Based on cardinal molecular features of the respective COO subtypes and applying the linear predictor score (LPS) classification method presented by the Staudt group for DLBCL[42], we developed a 21-gene murine version of the human 25-gene classifier, which was instrumental to recognize distinct Eμ-myc lymphoma subtypes, i.e., a GCB-like, an ABC-like and a third, unclassified group with ABC/GCB-overlapping features (Supplementary Fig. 1c). Importantly, TTR stratified by the GCB-like vs. ABC-like COO status of SENS lymphomas demonstrated a significantly superior long-term outcome to CTX in the GCB-like arm, thereby recapitulating the predictive role of the COO status in human DLBCL[8] (Fig. 1d). Thus, Myc-driven mouse lymphomas comprise biological features that resemble human GCB- and ABC-type DLBCL, in which high-level Myc expression is also a prominent and widely detectable feature[13,43].

Additional transcriptional DLBCL heterogeneity with associated distinct functional biological properties, albeit not necessarily different outcomes to standard therapy, was captured in the "comprehensive consensus cluster (CCC)" classification by the Shipp group[44]. Hence, we asked whether the CCC subtypes "B-cell receptor (BCR)/proliferation" signaling, "oxidative phosphorylation (OxPhos)" metabolism, and "host response (HR)" immunology would also be applicable to primary Eμ-myc lymphomas, separating them into these three distinct subclasses as well. Using a three-class extension of the LPS method on the Eμ-myc GEP data, this was indeed the case (Supplementary Fig. 1d). Remarkably, CCC class-specific upregulated gene sets, selected by GSEA in human DLBCL comparing each subgroup against the other two, were similarly enriched in the Eμ-myc model (Fig. 1e; DLBCL vs. Eμ-myc patterns in each of the three CCC subtypes). In essence, Eμ-myc transgenic lymphomas are molecularly heterogeneous and share prominent biological features related to distinct subclasses of human DLBCL, suggesting their suitability as a faithful functional model for certain features of the corresponding human disease.

**Molecular data possess predictive power across species.** We further sought to determine whether a strictly outcome-based (i.e., COO- or CCC-agnostic) DLBCL signature might also be predictive for Eμ-myc lymphomas and vice versa. We first analyzed GEP data from our 39 primary SENS Eμ-myc lymphomas, and utilized the most strongly differentially expressed genes to separate curable (i.e., NR) from relapsing (i.e., RP) lymphomas (Supplementary Fig. 2a). A similar approach was used to distinguish at diagnosis DLBCL patients that achieved lasting

remissions from those failing therapy (Supplementary Fig. 2b). Next, we selected genes which were commonly upregulated in matching response groups (i.e., progressors vs. nonprogressors) in both species. These genes were subsequently applied to the outcome-annotated expression data of murine and human lymphomas, and two top-clusters (cluster 1 and 2) were identified in each species by hierarchical cluster analysis (Fig. 2a, b; grey and black bars). These clusters were significantly correlated with the respective response groups (Fig. 2a, b; colored bars and Supplementary Fig. 2c, d). Strikingly, the two clusters sharply separated both CTX-exposed Eμ-myc lymphomas for significantly different TTR (Fig. 2c), and R-CHOP-treated DLBCL patients regarding their progression-free and overall survival (PFS and OS, respectively; Fig. 2d). Hence, individual transcriptome data linked to clinical response information can be utilized to generate gene stratifiers that discriminate good from poor risk lymphomas across species. However, transcriptome-based data not linked to functional biological insights may not suffice to decipher the contribution of distinct treatment effector principles, especially cellular senescence, in long-term outcome after anticancer therapies.

**Probing drug-induced senescence in the Eμ-myc lymphoma model.** Accordingly, we seek here to specifically address senescence as a drug-evoked response program, to link its genetic underpinnings to transcriptome-informed outcome data in murine and human lymphoma settings, and to develop predictors of the senescence contribution to patient survival. Starting from a mouse model platform with defined senescence-compromising gene defects, primary Eμ-myc transgenic LC, stably over-expressing Bcl2 (as naturally detected at high levels in many lymphomas including DLBCL[15]) to block drug-induced apoptosis, enter TIS in vitro and in vivo, if senescence-essential gene loci such as Suv39h1 or p53 alleles were not deleted[22,25,28] (Fig. 3a, Supplementary Fig. 3a, b). When investigated in a trial-like fashion in the absence of an intact apoptotic response, we observed a dramatically shortened OS of the mouse cohort bearing senescence-incapable Suv39h1-deficient (hereafter referred to as Suv39h1−) lymphomas, thereby suggesting a critical role for TIS in the long-term outcome to therapy (Fig. 3b).

Next, we asked whether senescence would also affect outcome when Bcl2 is not exogenously overexpressed. Importantly, non-Bcl2-engineered Suv39h1− LC presented—like control lymphomas, but different from apoptosis-compromised p53null lymphomas—with exquisite apoptotic drug sensitivity, thus highlighting their selective senescence defect while apoptosis remains intact (Supplementary Fig. 3c). Accordingly, mice harboring Suv39h1-deficient or proficient lymphomas typically entered a clinical CR in response to a single administration of CTX in vivo (i.e., all mice presented tumor-free at time-point 0 post CTX; Fig. 3c). Interestingly, when we examined three lymphoma compartments—the bone marrow (BM), the LN and the spleen (SP)—with respect to potentially remaining LC by Eμ-myc transgene-specific polymerase chain reaction (PCR) in a "minimal residual disease (MRD)"-like analysis 10 days after CTX, we found most of the control lymphoma mice-derived samples still positive, while none of the probes of the Suv39h1− group produced an MRD signal (Fig. 3d). Spleen sections were obtained to visualize remaining LC in situ, and unveiled much smaller islands of persistent cells in the Suv39h1− cohort (Fig. 3e). Though, the vast majority of these cells scored Ki67+/H3K9me3− by co-immunostaining—in stark contrast to the virtually Ki67-negative but H3K9me3-positive, hence, senescent splenic lymphoma residues in the control group, indicating not only a massive drop in overall tumor burden, but an important

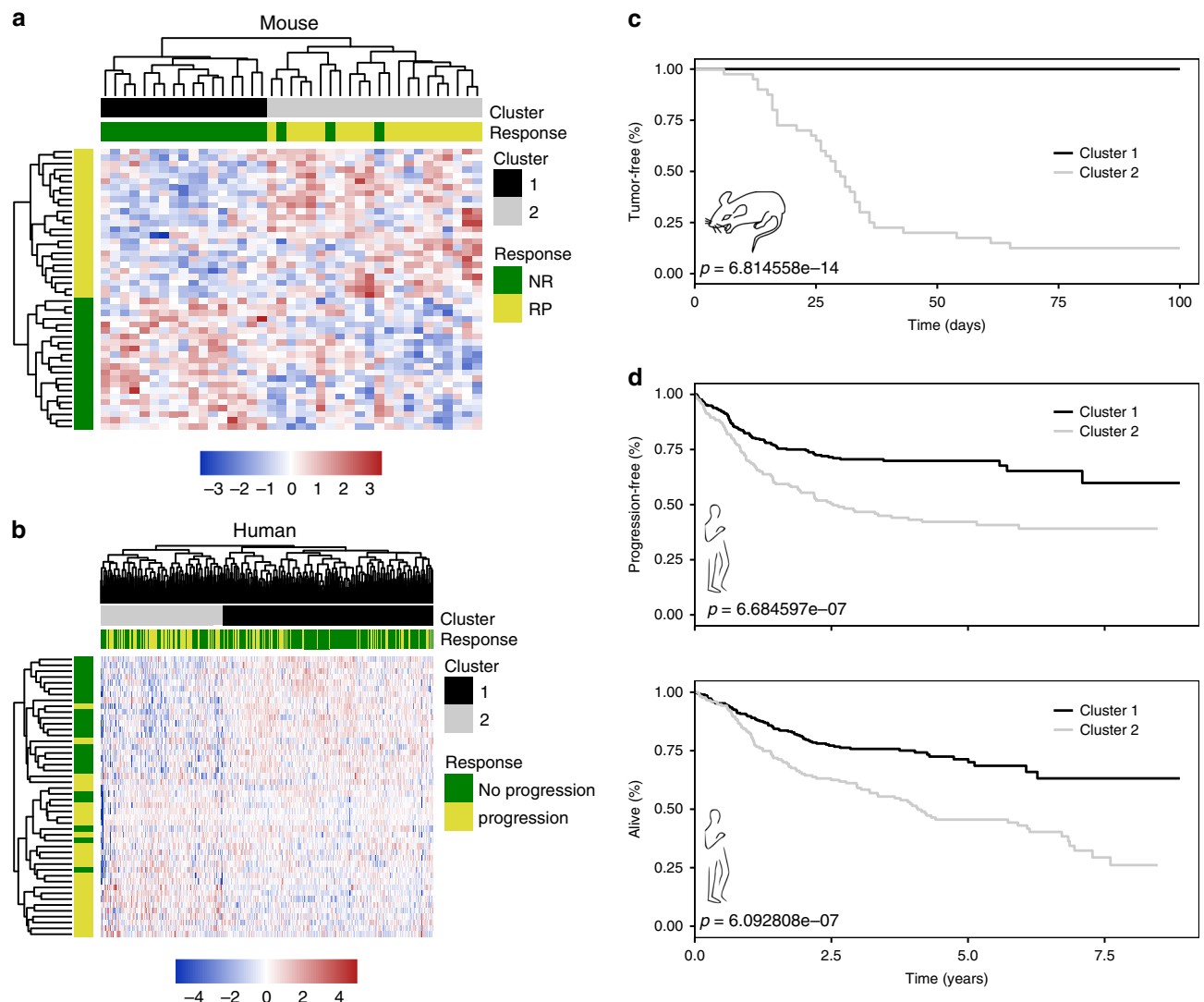

**Fig. 2 DLBCL and Eμ-*myc* lymphoma share predictive genes sets. a** Heatmap of 39 primary Eμ-*myc* lymphomas based on genes being commonly differentially expressed in non-progressing vs. relapsing DLBCL patients. **b** Same as in (**a**), but for 470 DLBCL patients (GSE31312). Eμ-*myc* lymphomas and DLBCL patients were separated into two clusters by hierarchical clustering using Pearson's distance correlation and complete linkage. Because of little very high and very low z-scores, they were restricted to the interval [−5,5] after hierarchical clustering for presentation purposes only. **c** Tumor-free survival of CTX-treated Eμ-*myc* lymphomas stratified by two hierarchical clusters ($n_{cluster\ 1} = 34$, $n_{cluster\ 2} = 42$) as shown in (**a**). Two mice per lymphoma were used. **d** Progression-free and overall survival of R-CHOP-treated DLBCL patients stratified by the two hierarchical clusters as shown in (**b**) ($n_{cluster\ 1} = 297$, $n_{cluster\ 2} = 173$).

qualitative state switch of the remaining Suv39h1-proficient LC (Fig. 3e). Matching the day-10 MRD data, mice of the Suv39h1⁻ group scanned by whole-body luciferase imaging regarding their tumor burden presented with virtually no remaining lymphoma load, while control lymphoma-bearing mice produced positive signals at this time-point (Fig. 3f and Supplementary Fig. 3d). However, as shown for a day-30 comparison, Suv39h1⁻ lymphomas rapidly progressed out of this complete molecular response (i.e., below-detectability) situation, whereas mice of the control group rather turned, with slow kinetics, luminescence imaging-negative (Fig. 3f, Supplementary Fig. 3d), thereby pointing toward a potential clinical pitfall when utilizing MRD or imaging diagnostics in a certain time slot of a senescence-dominated residual tumor load (further in line with the previously reported and equally unanticipated strong glucose avidity of robustly growth-terminated senescent lymphoma-bearing mice by positron emission tomography[25]). The data also imply a competition between the execution of apoptosis or

senescence on the cellular level of control tumors, whose nature might be stochastic or controlled by yet to-be-determined regulators. Importantly, when we compared clinical outcome in the absence of an apoptotic block, mice bearing Suv39h1-deficient lymphomas had a much shorter TTR and a much higher fraction of lymphomas that relapsed within the observation period of 100 days (Fig. 3c). Thus, we conclude that cellular senescence critically contributes to the long-term outcome after chemotherapy in both apoptotically compromised and competent lymphoma settings.

**Cross-species investigation of the H3K9 senescence relay.** While the histone methyltransferase Suv39h1 operates as an essential mediator of OIS and TIS in mouse lymphoid cells[23,25,28,34], other H3K9-active methyltransferases may compensate for loss of this moiety in different tissues and species. Given the central role of the H3K9me3 mark in senescence, we

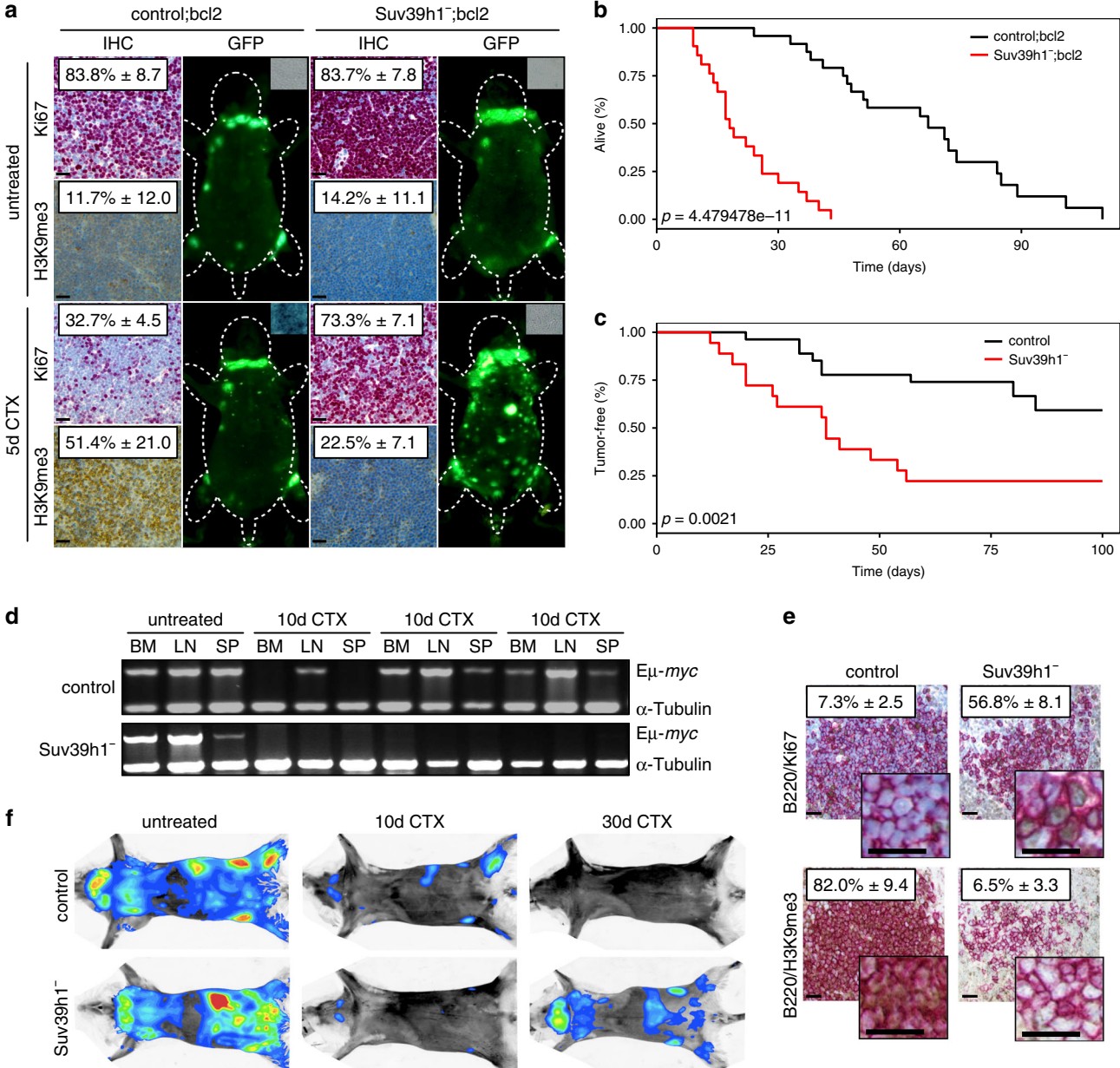

**Fig. 3 Defective senescence but intact apoptosis accounts for temporarily deeper responses of Suv39h1-deficient lymphomas. a** Ki67, SA-β-gal activity and H3K9me3 IHC analyses of lymphoma sections, plus whole-body fluorescence scans of mice bearing GFP-tagged control;bcl2 or Suv39h1⁻;bcl2 lymphomas. Results show mean percentages of cells positive for the respective markers in situ after a 5-day CTX exposure or those that remained untreated ± s.d. ($n = 3$ technical replicates per group). **b** Overall survival of mice bearing Suv39h1⁻;bcl2 ($n = 21$, red line) or control;bcl2 ($n = 24$, black line) lymphomas after CTX treatment. **c** Tumor-free survival (i.e., TTR) of mice bearing Suv39h1⁻ ($n = 18$, red line) or control ($n = 27$, black line) lymphomas after CTX treatment. **d** Eμ-*myc* transgene-specific PCR in Suv39h1⁻ and control Eμ-*myc* lymphomas prepared from bone marrow (BM), lymph-node (LN), or spleen (SP) from untreated mice (as a positive control) and 10 days after CTX treatment. Shown are analyses in mice bearing $n = 8$ individual lymphomas. Original photomicrographs are presented in Supplementary Fig. 7. **e** Spleen sections stained for Ki67 and H3K9 in B220-positive lymphomas from Suv39h1⁻ or control mice. Numbers indicate mean percentages of cells positive for the respective marker in situ 10 days after CTX administration ± s.d. (shown are representative photomicrographs of $n = 3$ biological replicates per group). **f** Representative whole-body luciferase imaging of mice harboring Suv39h1⁻ vs. control lymphoma (not bcl2-engineered) before treatment as well as 10 and 30 days after CTX treatment ($n = 3$ biological replicates per group [see also Supplementary Fig. 3d]). All scale bars in this figure represent 50 μm.

therefore asked, in turn, whether overexpression of two structurally unrelated H3K9me3-active demethylases, namely LSD1 (a.k.a. AOF2 or KDM1A) and JMJD2C (a.k.a. GASC1 or KDM4C), which we recently found to cancel OIS in melanomagenesis[45], and which were detected at elevated levels in numerous tumor entities[46–51], might also counter TIS. Stable transfer of LSD1 and JMJD2C genes into control;bcl2 lymphomas, similar to

inactivation of Suv39h1, resulted in sharply reduced levels of the H3K9me3 mark and the senescence-associated cyclin-dependent kinase inhibitor p16^INK4a by immunoblot analysis, and impaired ADR-induced senescence in vitro, while leaving drug-induced DNA damage response signaling via γ-H2AX and serine-18-phosphorylated p53 (p53-P-Ser18) intact (Fig. 4a, Supplementary Fig. 4a), resulting in demethylase-unaffected apoptotic death in

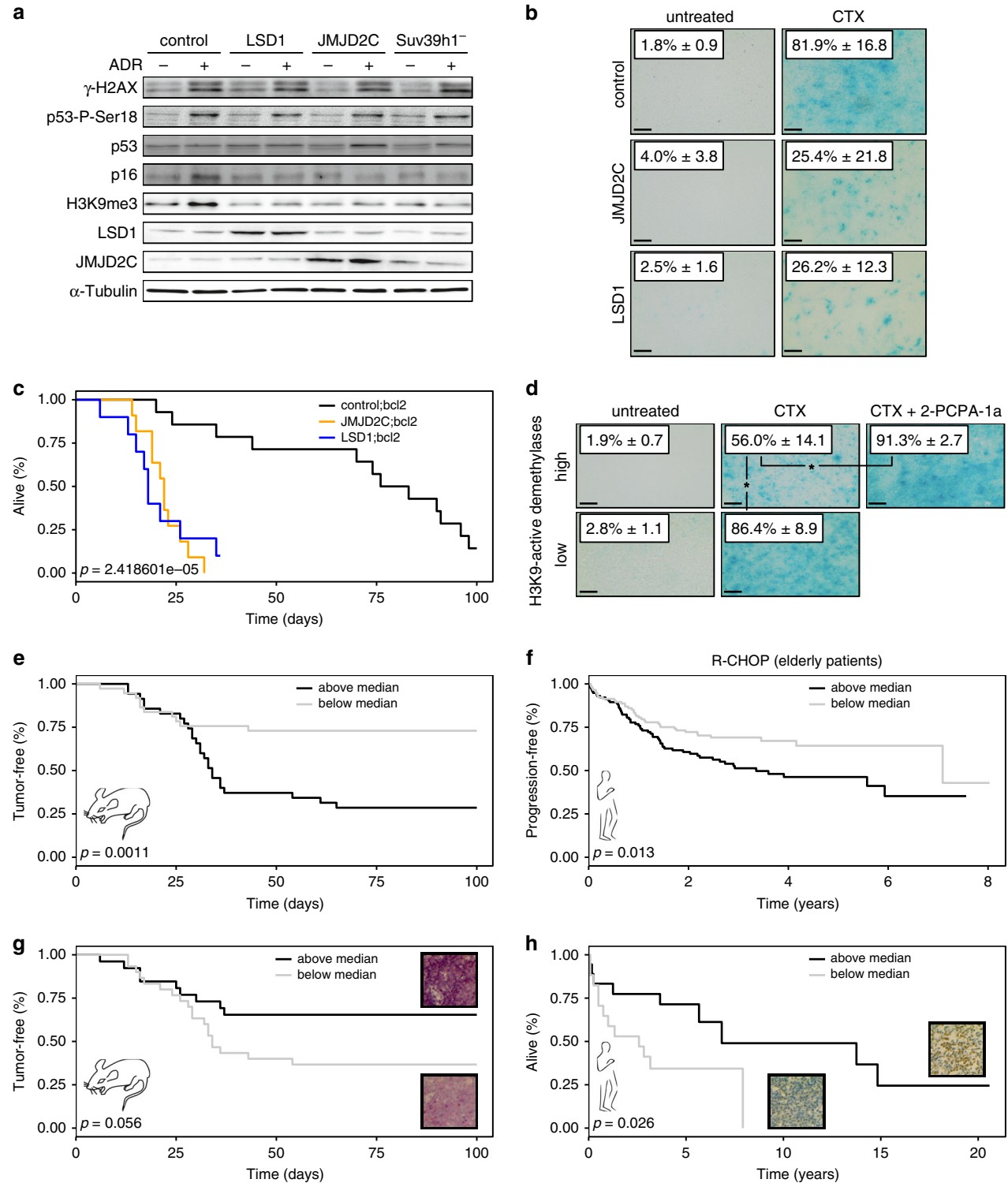

non-Bcl2-protected lymphomas (Supplementary Fig. 4b). As observed for mice bearing *Suv39h1⁻;bcl2* lymphomas, those carrying control;bcl2 lymphomas engineered to stably over-express either LSD1 or JMJD2C presented with very modest senescence induction, no significant enhancement of the H3K9me3-positive cell fraction, and a dramatically shortened OS after CTX therapy when compared to the respective vector control group, thus further underscoring the importance of an effective H3K9me3-governed senescence response for long-term disease control (Fig. 4b, c, Supplementary Fig. 4c). Moreover, SA-

β-gal activity reached much higher levels in ADR-exposed LSD1; bcl2 lymphomas if co-treated with the LSD1 inhibitor 2-PCPA-1a in vitro (Supplementary Fig. 4d), similarly to the high SA-β-gal activity induced by CTX exposure under 2-PCPA-1a co-treatment in vivo (Fig. 4d).

Next, we tested whether treatment responsiveness of Myc-driven control lymphomas that formed in the absence of any engineered H3K9me3-erasing moiety would relate to their endogenous H3K9 demethylase expression status. To investigate this, we focused on genes with H3K9-specific histone demethylase

**Fig. 4 The H3K9me3 mark operates as a central, outcome-relevant senescence relay. a** Western blot analysis of senescence-related proteins in Bcl2-protected LSD1- or JMJD2C-overexpressing, Suv39h1-deficient and control lymphomas after 5 days of ADR treatment in vitro or left untreated. Samples are representative of each genotype ($n > 3$). Original photomicrographs are presented in Supplementary Fig. 8. **b** Lymphoma sections of CTX-treated mice bearing Bcl2-protected JMJD2C- or LSD1-overexpressing vs. control lymphomas stained for SA-β-gal. Numbers indicate mean percentages of cells positive ± s.d. (shown are representative photomicrographs of biological replicates, $n = 4$ primary lymphomas for control;untreated, $n = 5$ for control;CTX, $n = 3$ for LSD1;untreated, $n = 4$ for LSD1;CTX, $n = 4$ for JMJD2C;untreated, $n = 4$ for JMJD2C;CTX). **c** Overall survival (OS) in CTX-treated mice bearing Bcl2-protected JMJD2C ($n = 11$, orange line)-, LSD1 ($n = 10$, blue line)-overexpressing or control ($n = 14$, black line) lymphomas. **d** Tumor sections of CTX- or CTX-plus-2-PCPA-1a-treated mice bearing lymphomas with low or high H3K9-active demethylases (GO:0032454) expression levels stained for SA-β-gal. Numbers indicate mean percentages of cells positive ± s.d. (shown are representative photomicrographs of $n = 3$ biological replicates per group). "*" represents statistical significance by unpaired t test, $p = 0.0129$ (high;CTX vs. high;CTX + 2-PCPA-1a) or $p = 0.0339$ (high;CTX vs. low;CTX). **e** Tumor-free survival of CTX-treated mice stratified by median expression of genes that belong to the GO term "histone demethylase activity (H3-K9 specific)" (GO:0032454). Two mice per lymphoma were used. Above median: $n = 37$ (black line), below median: $n = 39$ (grey line). **f** Progression-free survival (PFS) of elderly (>63 years; with their aged and presumably more senescence-prone lymphoma cells) R-CHOP-treated DLBCL patients (GSE31312) stratified as in (**e**). Above median: $n = 116$ (black line), below median: $n = 117$ (grey line). **g** Tumor-free survival of CTX-treated mice bearing Eμ-myc lymphomas with high ($n = 28$, black line) or low ($n = 31$, grey line) levels of the H3K9me3 mark. **h** OS of R-CHOP-treated DLBCL patients bearing lymphoma with high ($n = 18$, black line) or low ($n = 17$, grey line) levels of the H3K9me3 mark. All scale bars in this figure represent 100 μm.

activity (GO:0032454). Indeed, we found TIS much weaker in control lymphomas with globally high expression levels of H3K9-active demethylases, while the senescence response, and, accordingly, the fraction of H3K9me3-positive cells detected by immunostaining could be restored to an extent otherwise seen in low-level H3K9 demethylase expressers by treatment with the LSD1 inhibitor 2-PCPA-1a in vivo (Fig. 4d, Supplementary Fig. 4e). Moreover, a high endogenous H3K9 demethylase status, among others composed of LSD1 and JMJD2C expression levels, not only predicted senescence restoring-susceptibility to the LSD1 inhibitor 2-PCPA-1a but equally to the JMJD2 inhibitor IOX1, while neither inhibitor enhanced ADR-inducible senescence in lymphomas with low endogenous H3K9me3 demethylase status (Supplementary Fig. 4f). Notably, mice harboring individual lymphomas with global H3K9 demethylases expression above median presented with a significantly shorter TTR (Fig. 4e). Remarkably, when probing the transcriptome datasets of DLBCL patients for the humanized version of this H3K9 demethylase stratifier, a significantly shorter PFS was unveiled in elderly DLBCL patients if H3K9 demethylases were expressed above median in their lymphoma samples (Fig. 4f). Since we expected the activities of LSD1, JMJD2C and related demethylases to converge at the H3K9me3 mark, we explored the H3K9me3 status in situ as a putative biomarker of long-term outcome to treatment. Indeed, both the Eμ-myc cohort and DLBCL patients with high H3K9me3 expression in their lymphoma sections exhibited a significantly better long-term outcome to therapy when compared to those with H3K9me3 detectability below median (Fig. 4g, h). Hence, our data highlight TIS as an H3K9me3-governed outcome-improving drug effector mechanism first discovered and functionally dissected in murine lymphomas, and subsequently confirmed in human large B-cell lymphomas. Moreover, the endogenous H3K9 demethylase genetics of DLBCL imply that different baseline expression levels of the H3K9me3 mark as a readout of alterations acquired during lymphomagenesis prior to any drug encounter may serve as a predictive biomarker of long-term outcome to therapy.

**Senescence impairment plays a role in treatment failure.** To approach our ultimate goal of a TIS-focused genetic predictor in lymphoma, we defined an in vitro-senescence signature based on genes being differentially expressed between ADR-senescent control;bcl2 vs. equally ADR-exposed Suv39h1−;bcl2 and, hence, senescence-incapable, or untreated Bcl2-infected lymphomas (Fig. 5a). Next, we probed whether RES lymphomas typically present with a molecular senescence defect by applying the in vitro-senescence signature to our clinical trial-like mouse cohort

as an approximation to their senescence susceptibility (Fig. 5b; using the LPS classifier method and the in vitro-TIS dataset (Fig. 5a) as training data). Although TIS manifests with slow kinetics as a full-featured biological condition in response to DNA-damaging therapy, we hypothesized that early, albeit subtle senescence-reminiscent molecular changes might already become detectable within hours after drug exposure. Indeed, GSEA highlighted the proximity of 4-h-CTX-in vivo-challenged senescence-capable Eμ-myc lymphomas without Bcl2 protection and 5d-ADR-in vitro-senescent bcl2-engineered lymphomas based on similar ES of a large number of common gene sets in their expression profiles (Fig. 5c). If TIS contributes to superior long-term outcome, we further suspected that senescence inducibility might be more profound in curable lymphomas at diagnosis and selected against in relapsed lymphomas. Hierarchical cluster analysis from our clinical trial-like mouse model based on 4-hour-CTX administration in vivo further supported our hypothesis that RES lymphomas were predicted to have a senescence-incapable phenotype (Supplementary Fig. 5; note virtually no overlapping red- and black-labeled cases). Importantly, when applying the LPS classifier as a TIS score, we found RES lymphomas, unlike the SENS lymphoma groups, to be strongly skewed toward a Suv39h1-deficient, hence, senescence-defective pattern in response to the 4-h CTX challenge in vivo (Fig. 5d). Strikingly, when we next classified each individual lymphoma in the CTX-exposed condition as either a "senescence responder" (TIS score > 0.8), a "senescence non-responder" (TIS score < 0.2) or "unclassifiable" ($0.8 \leq$ TIS score $\geq 0.2$), the LPS/TIS score unveiled a significantly superior long-term outcome (TTR) for yet-to-be-treated lymphomas belonging to the senescence responder group (Fig. 5e). Moreover, our cross-species strategy also linked the GCB subtype of DLBCL samples at diagnosis, i.e. prior to any drug encounter, to a set of TIS-upregulated genes, thereby matching our previous experimental observation of a particularly strong TIS capacity in GCB-like and Bcl2-overexpressing, but not in ABC-like Eμ-myc lymphomas with their enrichment of "TIS down-regulated" and proliferation-associated gene sets[27] (Fig. 5f). Intriguingly, the more senescence-prone GCB subtype also appeared to be transcriptionally skewed toward an adult tissue stem cell (ATSC) signature in both murine and human lymphomas, thereby expanding on our recently reported finding on senescence-associated stemness in DLBCL[28]. Thus, early molecular changes detectable in response to CTX treatment seem to anticipate a later biologically fully established senescence response.

**"SUVARness" predicts long-term outcome in mice and men.** Finally, we sought to utilize senescence-related transcript

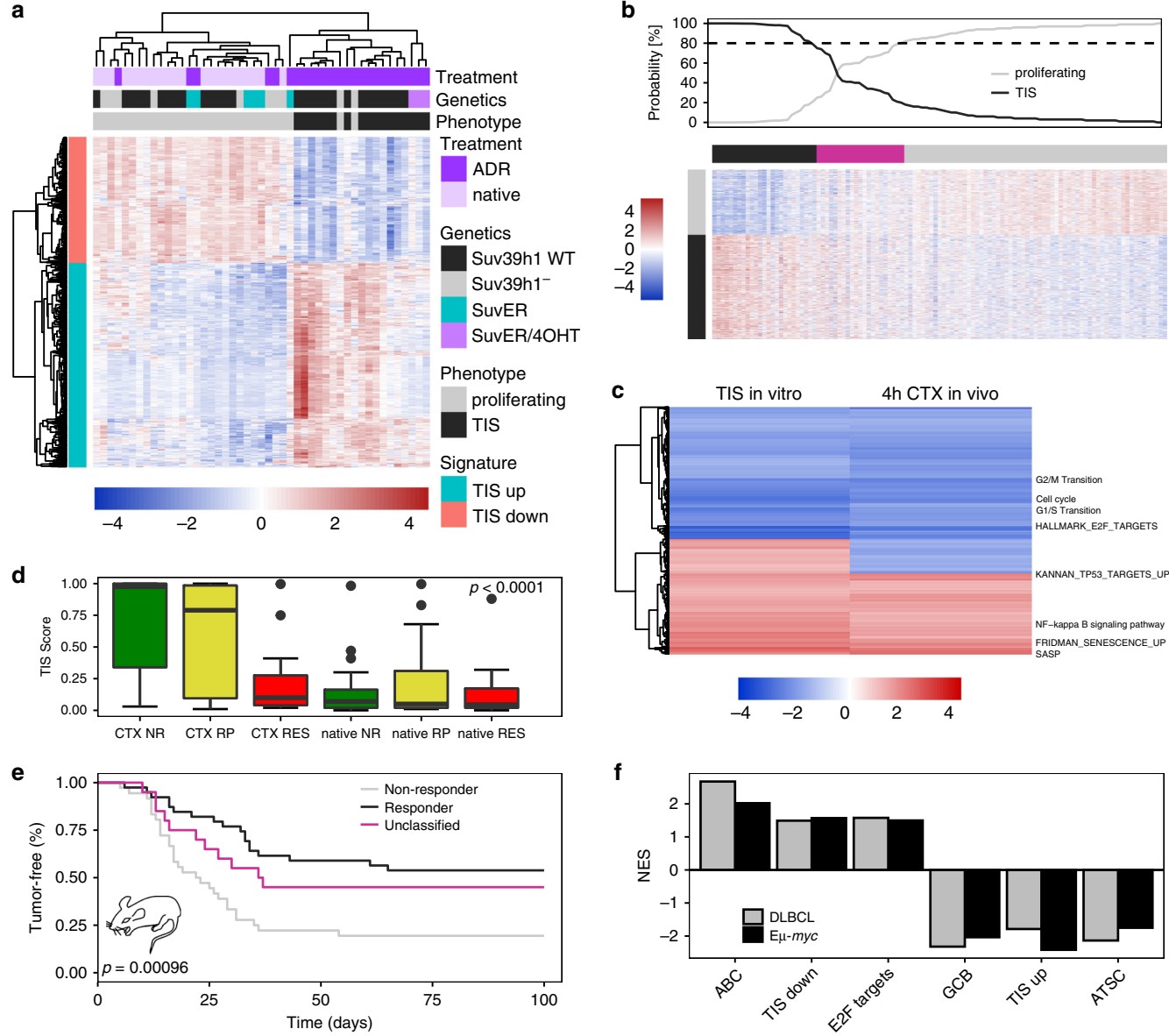

**Fig. 5 Senescence defects account for treatment failure. a** Heatmap of differentially expressed genes (p value < 0.01, log-FC > 1) comparing in vitro-ADR-treated control;bcl2 or ADR/4OHT-double-treated Suv39h1:ER;bcl2 senescent (TIS) to ADR-treated Suv39h1⁻ or Suv39h1:ER;bcl2 senescence-incapable, or untreated non-senescent lymphomas (collectively 47 lymphomas). **b** Heatmap and associated probability of TIS (black line) vs. no senescence (i.e., proliferating, grey line) in CTX-treated lymphomas based on LPS classification using data in (**a**), which had 85.1% overall accuracy in a tenfold cross validation of the in vitro-training data. **c** Heatmap of GSEA Normalized Enrichment Scores (NES) compiled of overall 47 5d ADR-in vitro-treated senescent control;bcl2 lymphomas vs. non-senescent untreated and Suv39h1⁻;bcl2 or non-4OHT-exposed Suv39h1:ER;bcl2 senescence-incapable lymphomas compared to 54 4 h-in vivo-CTX-treated vs. 55 untreated lymphomas. Clustering was performed using Euclidean distance. Senescence-related gene sets are shown, some of them specifically indicated[70]. **d** Probability of a TIS phenotype according to the LPS classification (cf. panel **b**, black line) of 4 h-CTX-treated or untreated lymphomas belonging to the indicated response groups (n = 20 CTX NR, n = 19 CTX RP, n = 15 CTX RES, n = 20 native NR, n = 19 native RP, n = 16 native RES). Higher TIS scores represent a stronger senescence likelihood, while lower scores indicate a proliferating phenotype. Box-and-whisker plot showing the median (black line inside the box), upper and lower quartile (box edges), minima and maxima (whiskers), and outliers (dots). The p value was calculated using the Kruskal–Wallis rank-sum test. **e** Tumor-free survival (TTR) of mice bearing senescence responder (TIS score > 0.8; n = 42), non-responder (TIS score < 0.2; n = 38) or unclassified (0.8 ≤ TIS score ≥ 0.2; n = 22) Eμ-myc lymphomas as determined from the LPS classification (based on the TIS status in the 4 h-CTX-exposed condition [cf. panel **d**]). Two mice per lymphoma were used. **f** GSEA NES results comparing ABC-like to GCB-like subtypes in DLBCL patients (ABC/GCB training set from GSE10846, n = 150) and Eμ-myc lymphomas (dataset as used in Fig. 1, ABC/GCB classified, n = 95).

information obtained in untreated lymphoma samples as a tool to predict long-term outcome. Specifically, we generated a signature named "SUVARness" (in scientific proximity to the term "BRCAness" invented for BRCA1/2 wild-type tumors that share molecular features of BRCA-mutant malignancies[52], albeit composed here as a transcriptional signature), consisting of TIS upregulated genes, thus marking the conceptual opposite of the senescence-compromising Suv39h1 deficiency in lymphoma (Fig. 5a). We found this gene set strongly enriched in TIS control; bcl2 compared to proliferating mouse lymphomas in vitro

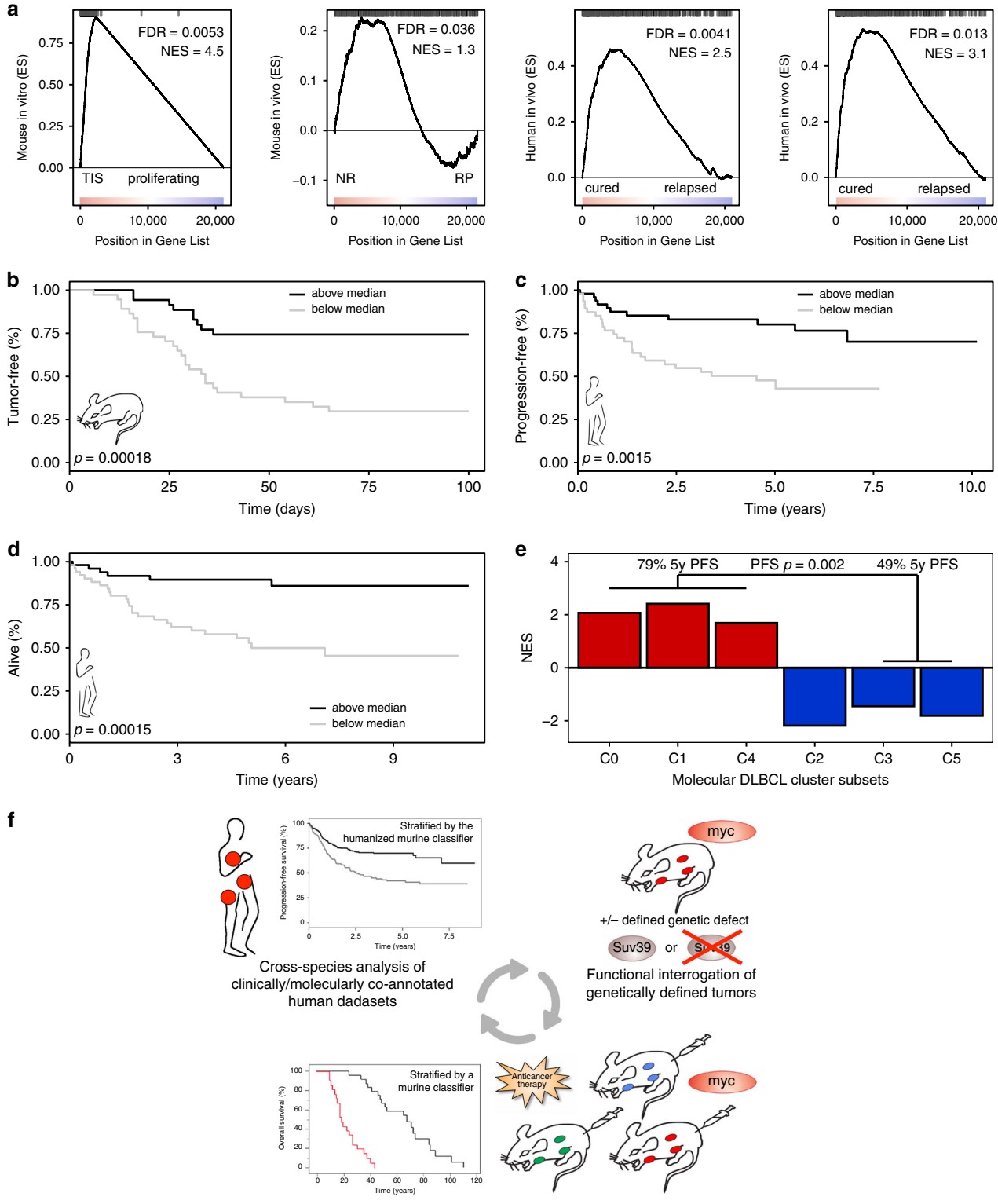

(Fig. 6a, first panel), and, likewise, strongly up in treatment-naïve NR Eμ-*myc* lymphomas in vivo (Fig. 6a, second panel), as well as in newly diagnosed DLBCL samples of two independent cohorts of patients who achieved lasting remissions, when being compared to RP mouse or human relapsed/refractory lymphomas, respectively (Fig. 6a, right panels). Based on these GSEA analyses, we defined a "core SUVARness" signature, commonly present in these GSEA's leading edges (i.e., the region from the top of the

gene list to the enrichment score [ES] profile peak position), consisting of 22, notably largely proliferation-independent and partly cell–cell/host immune interaction-relevant genes (Table S1), which might serve as a biomarker that anticipates treatment outcome (Fig. 6a). Accordingly, stratification of SENS Eμ-*myc* lymphoma based on the average expression of this core SUVARness signature predicted tumor-free survival, whereby mice bearing lymphomas with high-level expression experienced

**Fig. 6 The SUVARness gene signature is predictive for treatment outcome in DLBCL. a** GSEA profiles of TIS up-regulated (SUVARness) genes as shown in Fig. 5a probing Eμ-*myc*;bcl2 lymphomas TIS vs. untreated in vitro (outer left), SENS Eμ-*myc* lymphomas NR vs. RP (left), and DLBCL patients cured vs. relapsed in two cohorts (GSE31312, GSE98588; right). **b** Tumor-free survival in SENS Eμ-*myc* lymphomas stratified by average expression levels ($n = 38$ each arm) of the 22 core SUVARness genes (cf. Table S1) commonly selected from the GSEA leading edges (region from the top of the gene list to the enrichment score profile peak position) in (**a**). **c** Same as in (**b**), applying a fully humanized version of the core SUVARness classifier, showing PFS of R-CHOP-treated DLBCL patients (GSE98588). Above median: $n = 49$ (black line), below median: $n = 47$ (grey line). **d** OS for patients stratified as in (**c**). **e** GSEA results of core SUVARness-regulated genes in the recently reported newly defined molecular DLBCL cluster subtypes C0–C5 (ref. [14]), comparing each cluster vs. the others. According to Chapuy et al., patients in clusters C0/C1/C4, here marked as SUVARness-enriched, had a significantly better PFS compared to patients in clusters C3/C5 (ref. [14]). $n = 137$ ($n = 6$ C0, $n = 29$ C1, $n = 32$ C1, $n = 28$ C3, $n = 21$ C4, $n = 21$ C5). **f** Circular workflow of functional interrogations in mouse models of cancer harboring defined genetic lesions, investigation of genetically unmanipulated mouse tumors in a clinical trial-like setting in vivo, and cross-species application of the genetic determinants of novel biological functions and intervention-evoked dynamic state switches learned therein in corresponding human cancer patient cohorts.

superior survival (Fig. 6b). Cross-species application of a humanized version of the core SUVARness senescence classifier to the treatment-naïve transcriptome datasets of two independent human cohorts comprising several hundreds of R-CHOP-treated DLBCL patients also marked a much better PFS and OS of the higher-expressing half of these patients (Fig. 6c, d, Supplementary Fig. 6a, b). This was further validated in two independent CHOP-treated DLBCL cohorts (Supplementary Fig. 6c, d). In essence, our results highlight the power of the Eμ-*myc* transgenic mouse lymphoma model to unveil the importance of senescence, namely H3K9me3-governed SUVARness, in the course of human DLBCL treatment, equipping us with a molecular stratifier that unveils senescence susceptibility already at diagnosis in the patient sub-group that experiences significantly superior long-term outcome.

## Discussion

The molecular in-depth dissection of large sample numbers across most cancer entities in general and DLBCL in particular has led to a variety of genetically determined subgroups[6–15]. With the approach presented here, we add a biological and dynamic layer to the detailed but static profiling data obtained from patients at diagnosis. Demonstrating the genetic and clinical proximity of primary Eμ-*myc* transgenic mouse lymphomas to DLBCL provided the basis to further interrogate them with defined genetic defects in the biological feature of interest, or as unmanipulated samples in a clinical trial-like format, out of which we derived senescence-determining gene signatures whose humanized homologues proved predictive when applied to hundreds of DLBCL patients' data consisting of molecular profiles and the respective clinical outcome information post-standard therapy at resolution of the individual patient. Specifically, we were able to show that these lymphomas recapitulate genetic features of previously established DLBCL subtypes such as the HR/OxPhos/BCR consensus clusters and the clinically routinely utilized distinction by COO into ABC and GCB subtypes[6,42,44].

An important deliverable from this study is the humanized version of the core TIS upregulated gene set that links senescence susceptibility during subsequent therapy to improved long-term outcome in unselected DLBCL patients (cf. Fig. 6c, d, Supplementary Fig. 6a, b). We further applied this classifier of SUVARness, based on the original Suv39h1 liability of senescence[25,34], to very recently reported novel DLBCL subsets, termed clusters 0–5 (C0–C5), that were determined by a comprehensive in-depth genetic analysis of more than 300 primary DLBCL cases[14]. When probing individual clusters regarding their SUVARness, we found this outcome-predictive gene set to be specifically enriched for in C0, C1, and C4, while it was profoundly downregulated in C2, C5, and to a lesser extent in C3 (Fig. 6e). Thereby, SUVARness is linked to senescence-reminiscent features such as a SASP-like pro-inflammatory immune infiltrate (C0), enhanced NOTCH and NF-κB activity

(C1), and elevated RAS/BRAF/MEK signaling (C4), while *p53*, *INK4a/ARF*, or *Rb* inactivation reflect senescence-incapable genotypes in the SUVARness-negative cluster 2 (ref. [22]). Further senescence-focused investigations are needed regarding the role of MyD88–L265P mutations as one of the leading lesions in cluster 5 (C5), and the position of cluster C3, a strongly GCB subtype-enriched cluster with frequent *IgH* enhancer/*bcl2* trans-locations, the constellation for which we previously showed the NF-κB-hyperactive subset to be associated with senescence susceptibility[27]. Strikingly and further in support of the excellent outcome data presented here for various independent SUVARness-high DLBCL patient cohorts, grouped clusters 0, 1, and 4 were just reported to achieve a superior outcome (PFS) when compared to clusters C3/C5 (79% vs. 49% 5y-PFS, $p = 0.002$; ref. [14]).

Importantly, while this study pinpointed with SUVARness the ability to execute senescence as a tumor-controlling feature that accounts for improved long-term outcome to therapy when compared to tumors lacking such ability, TIS may also have detrimental implications. We recently discovered reprogramming of bulk tumor cells into cancer stem cells as the result of profound epigenetic remodeling in senescence[28], and reported that senescent cells may occasionally re-enter the cell-cycle by losing expression of senescence-essential gene moieties or gaining sufficient expression of senescence-countering gene activities[45]. Undoubtedly, senescence serves as a beneficial acute response to stresses that put tissue integrity at risk, but lastingly persistent senescent cells, for the reasons of pro-inflammatory SASP and potential senescence escape of SAS + tumor cells, represent a dangerous population one would rather like to selectively eliminate, if not cleared by host immunity anyway[53]. Consistently, we found here the particularly senescence-susceptible GCB subtype to present with features of an ATSC signature, underscoring the consideration of a "senolytic" approach at least in this subgroup. Since we reported the first successful, survival-prolonging demonstration of a senolytic strategy in cancer, specifically in the Eμ-*myc* lymphoma model, in vivo a few years ago[25], we are currently extending our DLBCL-focused studies to this question. In essence, TIS serves as an important back-up cell-cycle exit program especially in tumors where apoptosis is no longer available, but long-term persistence of senescent cells may evoke aggressive tumor properties if not effectively countered by a preventive senolytic intervention[32].

By introducing a primary transgenic lymphoma model with defined genetic lesions to interrogate distinct biologies—as presented here for the Suv39h1-dependent induction of cellular senescence in response to therapy—followed by the exploration of a genetically unbiased clinical trial-like mouse lymphoma cohort, all in a syngeneic, immune-competent context, we provide a DLBCL-approximated test platform for functional investigations. While a large body of literature underscores the successful

exploration of functional lymphoma features in Eμ-*myc* mice and the subsequent validation of these findings in human DLBCL[23,27,28,40,54–56], we certainly acknowledge limitations of this transgenic model as a reflection of DLBCL pathogenesis, particularly in light of numerous mouse models developed to more faithfully recapitulate GCB- or ABC-subtype features of human DLBCL[57–68]. However, while those models elegantly provide examples for distinct routes into GCB- or ABC-skewed diffuse large B-cell lymphomagenesis, they are, in turn, selectively composed of complexity-reduced genetics out of the overwhelming heterogeneity human DLBCL exhibit as a cardinal property. In contrast, initiated by a single *myc* transgene, it is probably exactly this broad spectrum of early to rather mature B-cell lymphomas and their diverse secondary hits which accounts for a heterogeneity in the Eμ-*myc* model that seems to recapitulate at least some critical features of the heterogeneous DLBCL transcriptomes[40,54]. Those resulting Eμ-*myc* transgenics-derived genetic determinants can be faithfully re-applied to human clinically and molecularly co-annotated datasets to identify hitherto unknown patient subgroups and to unveil the predictive implication of a certain biological response that a (sizeable) fraction of the newly diagnosed patients will experience in the course of their yet-to-start treatment (Fig. 6f). Of note, any gene candidate emerging from the human data analysis can be re-fueled into the system to launch a next translational cycle of functional studies in the mouse that will further deepen molecular insights into the tumor-treatment interplay, and eventually refine patient strategies in the clinic. Taken together, our approach highlights how functional mouse models inform decision making in cancer precision medicine beyond mutational analyses and expression profiling of the patient material.

## Methods

For the list of primers, details of microarray data processing and subsequent gene expression analysis, please see Supplementary methods.

**Mouse strains and lymphoma generation.** All animal protocols used in this study were approved by the governmental review board (Landesamt Berlin), and conform to the appropriate regulatory standards. All mice are in a C57BL/6 strain background. Eμ-*myc* transgenic lymphomas with defined genetic defects in the *suv39h1* or *p53* locus were generated by cross-breeding to *suv39h1* or *p53* knockout mice[23,25,28]. Genotyping and semiquantitative detection of the Eμ-*myc* transgene were carried by transgene-specific genomic PCR[20] and primers are listed in the Supplementary Information. Inducible Suv39h1 Myc-driven lymphomas were generated from Suv39h1⁻ Eμ-*myc* lymphomas stably transduced with a 4-hydroxy-Tamoxifen (4-OHT)-inducible Suv39h1/estrogen receptor fusion cDNA (Suv39h1:ER)[28].

**In vivo-treatments and mouse imaging.** Individual lymphomas were propagated in up to two strain-matched, non-transgenic (i.e., genetically non-engineered wild-type), fully immuno-competent 6–8-week-old mice each via tail vein injection of $10^6$ viable cells (notably, in rare cases, discordant NR/RP responses of the two same-lymphoma recipients may produce unequal medians of the stratified response groups). Recipient mice were treated with a single intraperitoneal dose of cyclophosphamide (CTX, Sigma, 300 mg/kg body weight) or CHOP (i.e., 150 mg/kg CTX, 3.3 mg/kg ADR, 0.5 mg/kg Vincristine, and 200 μg/kg Prednisone) at the time well-palpable LN enlargements (i.e., about 8–10 mm in diameter) had formed. Regarding the CHOP regimen, each drug was administered on day 1, followed by daily administration of Prednisone alone over the next 4 days, consistent with the schedule used in the clinic. Treatment responses were monitored during regular visits (at least twice a week; for a maximum of a 100-day observation period), which always included a general inspection of the mice and palpation of the typically affected LN regions (i.e. cervical, pre-scapular/axillar and inguinal) as time-to-relapse (TTR), i.e., progression-free survival (PFS), or reflect the time between treatment and unexpected death of the animal or a preterminal disease stage, collectively measured as overall survival (OS)[20]. After $CO_2$ euthanasia, LN, and in some experiments also the spleen or BM, were isolated to generate tissue sections or single-cell suspensions[20]. In vivo-animal imaging was performed as whole-body luciferase or GFP imaging using a LAS 4000 luminescence and fluorescence live imaging system (Fujifilm)[25].

**Cell culture, retroviral infection, and in vitro-treatments.** LC were cultured on irradiated NIH3T3 fibroblasts serving as feeder cells[20]. Retroviral gene transfer was carried out with infectious supernatant from murine stem cell retrovirus (MSCV)-transfected Phoenix packaging cells[24]. Full-length cDNAs of Lsd1 and Jmjd2c were purchased from GE Healthcare Dharmacon and subcloned into the MSCV backbone co-encoding a puromycin antibiotic resistance gene. Bcl2 was subcloned into MSCV co-expressing GFP. In vitro drug assays were performed using adriamycin (ADR), a topoisomerase II inhibitor, at a concentration of 0.05 μg/ml, unless otherwise indicated.

**Analysis of cell viability, cell-cycle, and senescence.** Viability and cell numbers were analyzed by trypan blue dye exclusion. Cell-cycle status was measured via incorporation of BrdU and PI labeling by flow cytometry[23,24]. Ki67 was visualized by immunohistochemistry as an indicator of cell proliferation, and SA-β-gal activity at pH 5.5 was assessed as a senescence marker in cytospin preparations or cryosections[22,34]. The JMJD2 inhibitor IOX1 came from Sigma, the LSD1 inhibitor 2-PCPA-1a was purchased from Uorsy, Ukraine (BBV27286962)[45].

**Immunoblotting and immunohistochemistry.** Immunoblotting (IB) and immunohistochemistry (IHC) were performed as described previously[20,27]. Antigen detection by IB was carried out with antibodies against γ-H2AX (Cell Signaling Technology [CST], #9718, 1:1000 dilution), p53-P-Ser18 (CST, #9284S, 1:1000), p53 (Leica Biosystems, #NCL-p53-CM5p, 1:1000), p16^INK4a (Santa Cruz Biotechnology, #sc-1661, 1:1000), H3K9me3 (Abcam, #ab8898, 1:1000), LSD1 (CST, #2184S, 1:1000), JMJD2C (Abcam, #ab85454, 1:1000) and α-Tubulin (Sigma, #T5168, 1:1000). Anti-mouse or anti-rabbit horseradish peroxidase-conjugated antibodies were used as secondary antibodies (GE Healthcare, #RPN4301, 1:1000 and #NA931V, 1:2000, respectively). Antigen detection by IHC was performed with antibodies against Ki67 (Dako, TEC3, 1:1000) and H3K9me3 (Abcam, #ab8898, 1:1000).

**Transcriptome profiling.** RNA was isolated from LC using the RNAeasy Mini kit (Qiagen) and hybridized to Affymetrix Mouse Gene 1.0 ST or Genome 430 2.0 microarrays according to the manufacturer's instructions. Arrays were hybridized, washed and scanned by standard Affymetrix protocols. Further processing of the raw microarray data is described below.

**Statistics and data analysis.** Based on previous experience with the Eμ-*myc* transgenic mouse lymphoma model, sample sizes typically reflect three to five, in some experiments also much higher numbers of individual primary tumors as biological replicates. For assessing long-term outcome after in vivo-treatments, seven or more tumor-bearing animals per arm were used. Survival analysis (Figs. 1b, d and 2c, d; 3b, c3; 4c and e–h; 5e; and 6b–d) was done using the *survival* package in R. Statistical significance of differences in the survival times were assessed using the log-rank test. All quantifications from staining reactions were carried out by an independent and blinded second examiner and reflect at least three samples with at least 100 events counted (typically in three different areas) each, and a *t* test was applied. Unless otherwise stated, a *p* value < 0.05 was considered statistically significant. For multiple testing corrections the method by Benjamini and Hochberg (BH) to control for false-discovery rate was applied[69]. Bioinformatics Analysis was performed in R 3.5.0 & Bioconductor 3.7 using various R packages as described in the Supplemental Experimental Procedures in greater detail. In order to ensure reproducible results, the R workspace was initiated with a random seed of 1.

**Reporting summary.** Further information on research design is available in the Nature Research Reporting Summary linked to this article.

## Data availability

The mouse model-derived raw microarray data—from our previously published control; bcl2, Suv39h1⁻;bcl2 and Suv39h1⁻;bcl2 transduced with 4OHT-inducible Suv39h1 (Suv39h1:ER;bcl2) lymphomas[25,28] – were deposited at the Gene Expression Omnibus (GEO) repository of the National Center for Biotechnology Information under accession number (GSE134753). Data from our clinical-trial like model were deposited under accession number (GSE134751). Expression data of 39 primary Eμ-*myc* lymphomas from our clinical trial-like model were combined with expression profiles from publicly available primary Eμ-*myc* lymphoma (https://www.ncbi.nlm.nih.gov/geo/query/acc.cgi?acc=GSE40760). Expression data of DLBCL patients are publicly available from NCBI GEO, comprising GSE10846, (GSE4475), (GSE4732) and (GSE31312). In addition, GEP of DLBCL patients from the Shipp lab for CCC DLBCL distinction were obtained from https://portals.broadinstitute.org/cgi-bin/cancer/publications/pub_paper.cgi?mode=view&paper_id=102 (ref. [44]) and for GSE98588 from NCBI GEO[14]. Raw CEL files were downloaded and processed using RMA implemented in the R package *oligo* and batch effects between scan dates reduced using ComBat in the *sva* package.

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

## Acknowledgements

We thank the late A. Harris, T. Jacks, and T. Jenuwein for mice, and members of the Schmitt lab for discussions and editorial advice. This work was supported by grants to J.K. and C.A.S. from the Deutsche Krebshilfe (No. 110678), to C.A.S from the BMBF CancerSys program project ProSiTu (No. 0316047C), the Helmholtz Association (within the "Preclinical Comprehensive Cancer [PCCC]" mouse model consortium; No. HA-305), the Deutsche Forschungsgemeinschaft DFG (GO 2688/1-1 | SCHM 1633/11-1 and SCHM 1633/9-1), and the Förderverein Hämatologie und internistische Onkologie (Tyle Private Foundation, Linz, Austria), and by a Clinical Scientist fellowship to J.K. from the Stiftung Charité and the Volkswagen Foundation. This interdisciplinary work was further made possible by the Berlin School of Integrative Oncology (BSIO) graduate program funded within the German Excellence Initiative, and the German Cancer Consortium (GCC). We acknowledge support from the German Research Foundation (DFG) and the Open Access Publication Funds of Charité–Universitätsmedizin Berlin.

## Author contributions

C.A.S. together with S.L and M.R. conceived the project, designed the experiments, interpreted the results and wrote the paper. J.K., J.R.D., A.B, E.M.W., D.N.Y.F., and M.M. conducted, assisted by A.L., functional mouse experiments including pharmacological inhibitor and chemotherapy studies in vitro and in vivo. P.L., D.L., and M.H. provided histo- and molecularpathological analyses. Y.Y. performed immunoblot analyses. B.C. probed DLBCL cluster datatsets. K.S., S.T., and U.L. compiled and processed raw data and carried out bioinformatic assays.

## Competing interests

The authors declare no competing interests.
