## [Peer Review File · Nature Communications]

Reviewers' comments:

Reviewer #1 (Remarks to the Author); expert in senescence and cancer, mouse models:

In this manuscript entitled "Mouse modelling of H3K9ME3-governed senescence predicts lymphoma patient outcome", Schleich et al. explore a transgenic mouse lymphoma model that recapitulates molecular features and pathological manifestations of patients diagnosed with diffuse large B-cell lymphoma (DLBCL). The authors interrogate the Eu-myc transgenic lymphoma capacity to induce chemotherapy-induced senescence and its epigenetic regulation by manipulating the methyltransferase Suv(ar)39h1 and H3K9me3-active demethylases in loss- and gain-of-function assays. Instrumental datasets from patients are employed to validate tumour progression, survival and the generated transcriptome profiling, which allows the authors gaining insight into the genetic mechanisms and clinical relevance of biological response programs. Remarkably, Schleich and colleagues propose a senescence-indicating gene signature that correlates with high-level H3K9me3 expression, termed "SUVARness", to predict favourable DLBCL patient outcome and to exploit cancer precision medicine in the clinic.

This work is interesting and delivers novel predictive tools for long-term outcome in lymphoma patients. The experiments are, generally, well designed and elegantly executed, providing a solid basis for the conclusions claimed. I have, however, some concerns that need further clarification.

Major concerns

1. The experiments performed in Figure 3d-f were done with low numbers of mice (n=3). This is particularly relevant for the result provided in Figure 3f, where a quantification of the whole-body luciferase imaging is not shown. Since the data appear to imply a competition between the execution of apoptotic and senescent programs, it would be ideal to include both markers of apoptosis and senescence in the experiments, which are missing, and to show data on the transcriptional profiling of apoptosis in subsequent figures. I find crucial to include p53null, control;bcl2 and Suv39h1-;bcl2 backgrounds in the experiment shown in Figure 3f in order to generate more compelling evidences to support the conclusion. Regarding Figure 3d, 30d CTX is missing, and sections of the different organs (not only the spleen) should be included in Figure 3e.
2. In the experiment shown in Figure 4a, why was it performed by using ADR treatment while the rest of the panels in the same figure refer to CTX? The use of ADR and CTX is sometimes confusing and it would be good to justify the chosen therapies.
3. Similarly to LSD1 and 2-PCPA-1a, could the results on JMJD2C be validated by using an appropriate inhibitor?

Minor points

1. Whereas the manuscript is well written, I find the Discussion a little bit short in its current state. This section would benefit from a deeper and more detailed analysis. Among other things, I think the authors should comment on their previous finding of senescence reversion and how it may impact their conclusions in this manuscript. Since therapy-induced senescence seems important for the long-term outcome, in particular the GCB subtype, would this finding preclude the use of senolytics as a potential combination therapy?
2. I guess that the statement in page 13 "Moreover, SA-B-gal activity reached much higher levels in ADR-exposes LSD1;bcl2 lymphomas if co-treated with the LSD1 inhibitor 2-PCPA-1a in vitro (Supplementary Fig 4d)" also refers to Figure 4d, please include in the text.

Reviewer #2 (Remarks to the Author); expert in lymphoma, gene signature, mouse models:

The manuscript by Schleich et al uses a MYC-driven lymphoma mouse model, in which tumor-induced senescence was shown to depend on the H3K9 histone methyltransferase Suv39h1, and combinations of it with genetic lesions of BCL2, TP53 and Suv39h1 to study treatment response to cyclophosphamide, a component of the R-CHOP regimen that is a gold standard for the treatment of DLBCL. The authors explore the role of H3K9 trimethylation and treatment-induced senescence as markers of treatment response and generate a signature of 'SUVARness' (senescence related transcripts induced by therapy) that is predictive of outcome in both EmuMYC lymphomas and human DLBCL samples.

Through a large amount of data and in line with their expertise in cellular senescence the authors provide interesting observations on the role of this process in therapy resistance. Unfortunately, the work is based on a wrong premise, that is the use of the E μ MYC mouse model as paradigm for human DLBCL. This is a significant flaw that undermines the relevance of the findings and the impact of the conclusions to the therapeutic management of DLBCL.

Since it was discovered over 20 years ago that DLBCL arises from germinal centre B cells, a large body of literature has established that tumors developing in E μ MYC mice have little to do with the human disease: EmMYC tumors are lymphoblastic B cell lymphomas and early B cell leukemias that arise from immature, pre-B cells or in a minority of cases from naïve B cells, and lack key genetic (somatic hypermutation of the immunoglobulin V genes) and phenotypic features (expression of GC markers like BCL6 and GL7) of GC descentance (eg Harris, J. Exp. Med. 1988; ; Sidman, Leukemia 1993). As such, the E μ MYC mouse can be useful to address certain questions related to MYC function, but does not recapitulate the cellular context of DLBCL and mature B cell lymphomas in general.

As the EmuMYC tumors are not GC derived, it is unclear how the authors could classify them in ABC-DLBCL and GCB-DLBCL (the data in Figure S1C are indeed weak). Mouse GC B cells and activated B cells should be used as a training set to define the linear predictor score, as done for the human tumors in Alizadeh et al 2001. Differences in NF- κ B transcriptional signatures may be reminiscent of certain aspects of ABC-DLBCL but they are not sufficient to classify a tumor into the ABC-subtype of DLBCL given that this pathway is engaged in a multitude of cancers.

The parallel made between the pattern of response/relapse to cyclophosphamide in the mouse model and the behavior of DLBCL patients is also misleading and cannot be taken as an evidence for the EmMYC tumors to "recapitulate treatment outcome of DLBCL". Cyclophosphamide is used in several regimens in addition to R-CHOP, and the high initial chemosensitivity with relapse in a subset of mice and progressively reduced response to secondary treatments is a feature of many experimental systems.

Statements such as "these lymphomas recapitulate clinical, histopathological and genetic features of human lymphoma" or are "faithful models of DLBCL" or have "genetic and clinical proximity to DLBCL" are incorrect and confusing.

The manuscript is difficult to read; limited use of adverbs and shorter sentences are recommended.

Point-by-point response (highlighted in blue) to the reviewers' comments

Reviewer #1 (Remarks to the Author); expert in senescence and cancer, mouse models:

In this manuscript entitled "Mouse modelling of H3K9ME3-governed senescence predicts lymphoma patient outcome", Schleich et al. explore a transgenic mouse lymphoma model that recapitulates molecular features and pathological manifestations of patients diagnosed with diffuse large B-cell lymphoma (DLBCL). The authors interrogate the Eu-myc transgenic lymphoma capacity to induce chemotherapy-induced senescence and its epigenetic regulation by manipulating the methyltransferase Suv(ar)39h1 and H3K9me3-active demethylases in loss- and gain-of-function assays. Instrumental datasets from patients are employed to validate tumour progression, survival and the generated transcriptome profiling, which allows the authors gaining insight into the genetic mechanisms and clinical relevance of biological response programs. Remarkably, Schleich and colleagues propose a senescence-indicating gene signature that correlates with high-level H3K9me3 expression, termed "SUVA_Rness", to predict favourable DLBCL patient outcome and to exploit cancer precision medicine in the clinic.

This work is interesting and delivers novel predictive tools for long-term outcome in lymphoma patients. The experiments are, generally, well designed and elegantly executed, providing a solid basis for the conclusions claimed. I have, however, some concerns that need further clarification.

Major concerns

1. The experiments performed in Figure 3d-f were done with low numbers of mice (n=3). This is particularly relevant for the result provided in Figure 3f, where a quantification of the whole-body luciferase imaging is not shown.

→ The experiments in Fig. 3d actually include eight, not three, mice (we apologize for the mistake and corrected the number in the revised version), with an investigation at three compartments in every animal, thereby generating 24 data points – which show high intra- and inter-animal consistency and a strong difference between the two comparator genotypes tested. Likewise, results in Fig. 3e mark clearly different means and reasonably low standard deviations, hence, reflecting both statistically significant, and, consistent with other experimental evidence provided here, biologically robust and meaningful findings. We appreciate the referee's suggestion to demonstrate reproducibility of the whole-body imaging data in Fig. 3f – and have now added an independent set of bioluminescence-imaged mice in Supplementary Fig. 3d of the revised manuscript. Actual quantification is difficult, since this analysis is not about similar tumor reduction/re-progression kinetics [which would allow a mean ± SDEV presentation], but rather the qualitative pattern of a somewhat reduced (day 10) to further reduced (day 30) tumor burden in mice carrying senescence-capable lymphomas ("control"), as compared to a profoundly reduced (day 10) tumor load that has already re-progressed shortly later (day 30) in mice bearing senescence-incapable lymphoma ("Suv39h1-/-"). Moreover, Fig. 3d and 3e provide semi-quantitative and quantitative assessments that complement Fig. 3f.

Since the data appear to imply a competition between the execution of apoptotic and senescent programs, it would be ideal to include both markers of apoptosis and senescence in the experiments, which are missing, and to show data on the transcriptional profiling of apoptosis in subsequent figures. I find crucial to include p53null, control;bcl2 and Suv39h1-;bcl2 backgrounds in the experiment shown in Figure 3f in order to generate more compelling evidences to support the conclusion.

→ The referee asks a great question, when suggesting to further analyze competition between apoptosis and senescence. We extensively characterized acutely drug-inducible apoptosis in non-bcl2-engineered settings as well as drug-inducible senescence in Bcl2-protected settings both *in vitro* and *in vivo* in the past (see, for example, Schmitt-CA et al., Genes Dev. 1999; Schmitt-CA et al., Cell 2002). In the non-bcl2-engineered condition, most lymphoma cells die within hours by apoptosis, as presented in Suppl. Fig. 3c in the current manuscript). Growth kinetics in heterogeneous populations are difficult to track (such as delayed apoptosis or senescence, or resumed proliferation of temporarily arrested cells); that is why we focused on detectability of senescence in a genotype-dependent manner (Suv39h1-proficient vs. -deficient) in population-based analyses at defined time-points *in vitro* and *in vivo*. Whether fate decisions between apoptosis vs. senescence are merely stochastic, or signaling network-governed, or the product of actual “cellular competition” is an intriguing question raised here by the referee that we are actively pursuing, with required experiments and techniques (*e.g.* single-cell RNA-seq, fate tracking, 3D-analysis of receptor/ligand interactions etc.) going well-beyond the scope of the current manuscript. The scientific point made in Fig. 3 is not about apoptosis *per se*; it is about the competing ability to enter senescence (instead of dying), thereby presenting with a higher residual tumor load, but “paradoxically” a much better long-term outcome as compared to a much deeper elimination of senescence-compromised but apoptosis-capable Suv39h1-deficient cells. In other words, the key question here is: how much tumor is left, and is the remainder largely senescent or not, if endogenous apoptosis has not been altered.

Fig. 3c-f critically complement Fig. 3a,b by now allowing the lymphoma cells to enter either apoptosis or senescence based on their natural genetic make-up, not guided by exogenous (apoptosis-blocking) Bcl2 overexpression. Any exogenous Bcl2 overexpression (in control, p53null or Suv39h1-deficient contexts), as suggested by the referee, would preclude us from looking into a strong cytoreductive response leading to a clinical complete remission with only “minimal residual disease (MRD)” lesions left (as presented). Outcome of Bcl2-protected Eμ-*myc* control vs. p53null or Suv39h1-deficient lymphomas is presented here (Fig. 3a,b), or was published elsewhere (Schmitt-CA et al., Cell 2002). Unlike Suv39h1 – controlling senescence, but not apoptosis (Suppl. Fig. 3c) – p53null lymphomas fail to enter both apoptosis and senescence (Suppl. Fig. S3c), thereby not allowing a senescence-focused interpretation. All genotypes engineered to overexpress Bcl2 can no longer achieve a remission; their lymph-nodes remain enlarged post-therapy due to arrest, senescence, or retained proliferative capacity; therefore, are

not suitable to study treatment-enforced MRD biology (*i.e.* the “senescent state switch”) in a close-to-the-clinic-fashion at a very low level-retained tumor burden.

Regarding Figure 3d, 30d CTX is missing, and sections of the different organs (not only the spleen) should be included in Figure 3e.

→ In the day-10 MRD analyses by lymphoma-specific Eμ-*myc* transgene PCR, we provide strong and discriminating evidence that control lymphomas remain MRD-positive, while Suv39h1-deficient lymphomas achieve MRD negativity. We agree with the referee (and said so in the text), that day 30 appears like a conversion of the findings, rendering control lymphomas largely negative by whole-body lymphoma scanning, while Suv39h1-deficient lymphomas already quantitatively re-appeared. Hence, MRD analyses were invariably negative for Suv39h1-deficient lymphomas at day 10 (consistent with no clinical re-progress at this early time-point according to Fig. 3c), whereas about 40% of these mice already clinically re-progressed at day 30 (Fig. 3c), hence, these and probably an additional proportion of mice relapsing within the next few days after day 30 can be expected to be positive for the lymphoma-specific Eμ-*myc* PCR at day 30, in line with a substantial burden of disease detectable by whole-body imaging (Fig. 3f, and the new Supplementary Fig. 3d). Given the long-term curability of approximately 50-60% of the control lymphomas in the absence of exogenous Bcl2 (Fig. 1b and 3c: around 75-90% of the mice remain in clinical remission at day 30), we expect day-30 MRD analyses in both lymphoma groups to be quite heterogeneous (and, hence, of no additional value beyond the data-rich Fig. 3c). While d30-MRD predictably is still negative or already positive in Suv39h1-deficient lymphoma-bearing mice, some control lymphoma-harboring mice will score positive for the MRD-PCR at this time, others become positive later, and the curable ones stay negative for good – thus, exhibiting huge heterogeneity, which does not add information beyond the detailed response analysis shown in Fig. 3c. The key point of Fig. 3 is the demonstration of an unexpected “paradox” pattern (as identified and shown for the day-10 time-point), where mice harboring lymphomas with clearly superior long-term outcome (*i.e.* control lymphomas) consistently present as MRD-positive, while those bearing lymphomas with clearly inferior long-term outcome (*i.e.* Suv39h1-deficient lymphomas) were consistently MRD-negative at this point.

In essence: we provide here for non-bcl2-engineered control vs. Suv39h1-deficient lymphoma-bearing mice four types of response assessment – (I) individual quantification of progression-free survival (PFS) for every animal enrolled in this investigation (Fig. 3c; $n = 45!$; $p = 0.0021$), (II) lymphoma-specific PCR of untreated vs. CTX-treated mice at day 10 in various compartments (Fig. 3d; $n = 8$; consistent results within and robustly different results between the genotypes tested), (III) multi-parameter *in situ* analysis of individual spleen sections at day 10 (Fig. 3e; $n = 6$; strong differences, statistically significant), and (IV) whole-body luciferase imaging at three different time-points, *i.e.* untreated, day 10 and day 30 (Fig. 3f; $n = 6$; second set of mice presented in Supplementary Fig. 3d).

We did not include histological investigations of the bone marrow and the lymph nodes (as tested by MRD-PCR in Fig. 3d) in Fig. 3e, since the spleen, in our opinion, is the most suitable and representative organ to identify remaining MRD lesions as small islands of malignant cells in serial sections – as compared to the low cellularity post-therapy in the bone marrow and the size-normalized, and, thus, very small lymph-nodes. Please note that lymphoma-originated luciferase signals were equally detectable at day 10 by bioluminescence imaging in control lymphoma-bearing mice at all sites, *i.e.* over the spleen, various lymph-nodes and the femora (*i.e.* bone marrow; Fig. 3f and Supplementary Fig. 3d). Moreover, our MRD (*i.e.* E μ -myc transgene PCR) data, fully consistent across the three compartments in 7/8 mice tested, further indicate that the spleen serves as a representative organ site.

Given the specific questions and concerns brought up by the referee with respect to Figure 3, we realized that our description may not have optimally explained the goal and conclusions from the experiments presented. In our view, the data shown are very clear, biologically meaningful and statistically very robust, but require better explanation of the senescence-attributed “MRD paradox”. We substantially rephrased this section of the main text for clarity.

2. In the experiment shown in Figure 4a, why was it performed by using ADR treatment while the rest of the panels in the same figure refer to CTX? The use of ADR and CTX is sometimes confusing and it would be good to justify the chosen therapies.

→ The short answer is: we probably confused the referee by not clearly stating that Fig. 4a, unlike the rest of the figure, reflects an *in vitro*-analysis, in which Adriamycin (ADR) is preferentially used. We apologize for the potential misunderstanding and have now clarified this point in the legend.

In greater detail: both therapies are justified since they represent standard DNA-damaging chemo agents (topoisomerase poison Adriamycin [ADR] and alkylating agent cyclophosphamide [CTX]) used to treat a broad spectrum of cancers in the clinic, and in combination as part of the lymphoma standard regimen CHOP, which – as a poly-agent regimen – has been applied in our study as well (C = CTX, H = ADR [see text and data in the context of Fig. 1 and 2, and Supplementary Fig. 1b in particular]). In the literature (including our own contributions), ADR has become one of the best-established chemo drugs to induce senescence *in vitro*, while single-agent ADR is less effective regarding tumor control and more toxic in mice when compared to CTX *in vivo*. CTX, however, is a prodrug that requires hepatic activation *in vivo*. Therefore, CTX cannot be used for *in vitro*-experiments. In our experience, senescence-related results obtained with either ADR or CTX *in vivo*, as well as ADR or the *in vitro*-active homologue of CTX, Mafosfamide, have been very similar – for the reasons, that the common underlying denominator is induction of DNA damage, subsequently evoking a DNA damage response. Additional datasets presented in the revised version of the manuscript (*e.g.* Supplementary Fig. 4f, compare to Fig. 4d) further underscore the comparability of ADR-*in vitro*- vs. CTX-*in vivo*-exposed settings.

3. Similarly to LSD1 and 2-PCPA-1a, could the results on JMJD2C be validated by using an appropriate inhibitor?

→ We are grateful for the referee's suggestion to validate LSD1/2-PCPA-1a findings similarly for IOX1, an inhibitor of JMJD family members (King-ON et al., PLoS One 2010), as we extensively did in a recent investigation of LSD1 and JMJD2C as structurally unrelated H3K9me3-active demethylases countering Ras/Braf-induced senescence and promoting melanomagenesis (Yu-Y et al., Cancer Cell 2018). Virtually all experiments using primary tumor material shown in this work demonstrated the interchangeability of 2-PCPA-1a and IOX1 regarding the functional restoration of cellular senescence in endogenous "H3K9 demethylase-high" tumors. We have now added – as novel Supplementary Fig. 4f – an analysis of lymphomas with high- vs. low-level endogenous H3K9 demethylase activity exposed to chemotherapy ± 2-PCPA-1a or ± IOX1, validating the susceptibility of demethylase-high lymphomas to either 2-PCPA-1a- or IOX1-mediated restoration of ADR-inducible senescence, and lack of such effect by both inhibitors in the H3K9 demethylase-low lymphoma samples.

Minor points

1. Whereas the manuscript is well written, I find the Discussion a little bit short in its current state. This section would benefit from a deeper and more detailed analysis. Among other things, I think the authors should comment on their previous finding of senescence reversion and how it may impact their conclusions in this manuscript. Since therapy-induced senescence seems important for the long-term outcome, in particular the GCB subtype, would this finding preclude the use of senolytics as a potential combination therapy?

→ The referee makes a number of excellent points here; hence, we followed the suggestion and expanded in the discussion on our recent findings on more detrimental features of persistent senescence – specifically regarding senescence-associated stemness (SAS; Milanovic-M et al., Nature 2018), and the not necessarily stable nature (to avoid the term "reversion", since escape from senescence is not equivalent with a reversibility to the pre-senescent state) of the senescent arrest (Yu-Y et al., Cancer Cell 2018). Since we linked SAS to the GCB subtype in Fig. 5f, we completely agree with the referee that this aspect should be addressed in the discussion. We added an extra paragraph devoted to this important point in the discussion of the revised manuscript, and specifically touched on the referee's question on the potential use of senolytics in certain DLBCL subtypes.

2. I guess that the statement in page 13 "Moreover, SA-B-gal activity reached much higher levels in

ADR-exposes LSD1;bcl2 lymphomas if co-treated with the LSD1 inhibitor 2-PCPA-1a in vitro (Supplementary Fig 4d)” also refers to Figure 4d, please include in the text.

→ Yes, the referee is right, and this demonstration of similar senescence effects ADR or CTX produce (as stated above in reply to “Major Concern 2”) in the presence or absence of an inhibitor (*i.e.* 2-PCPA-1a) should be more clearly emphasized. We are grateful for pointing this out and changed the text accordingly.

Reviewer #2 (Remarks to the Author); expert in lymphoma, gene signature, mouse models:

The manuscript by Schleich et al uses a MYC-driven lymphoma mouse model, in which tumor-induced senescence was shown to depend on the H3K9 histone methyltransferase Suv39h1, and combinations of it with genetic lesions of BCL2, TP53 and Suv39h1 to study treatment response to cyclophosphamide, a component of the R-CHOP regimen that is a gold standard for the treatment of DLBCL. The authors explore the role of H3K9 trimethylation and treatment-induced senescence as markers of treatment response and generate a signature of ‘SUVARness’ (senescence related transcripts induced by therapy) that is predictive of outcome in both EmuMYC lymphomas and human DLBCL samples.

Through a large amount of data and in line with their expertise in cellular senescence the authors provide interesting observations on the role of this process in therapy resistance.

→ We highly appreciate the judgement of the referee, who acknowledges our expertise and finds our observations, linking senescence to treatment outcome, interesting.

Unfortunately, the work is based on a wrong premise, that is the use of the E μ MYC mouse model as paradigm for human DLBCL. This is a significant flaw that undermines the relevance of the findings and the impact of the conclusions to the therapeutic management of DLBCL. Since it was discovered over 20 years ago that DLBCL arises from germinal centre B cells, a large body of literature has established that tumors developing in E μ MYC mice have little to do with the human disease: EmMYC tumors are lymphoblastic B cell lymphomas and early B cell leukemias that arise from immature, pre-B cells or in a minority of cases from naïve B cells, and lack key genetic (somatic hypermutation of the immunoglobulin V genes) and phenotypic features (expression of GC markers like BCL6 and GL7) of GC descentance (eg Harris, J. Exp. Med. 1988; ; Sidman, Leukemia 1993). As such, the E μ MYC mouse can be useful to address certain questions related to MYC function, but does not recapitulate the cellular context of DLBCL and mature B cell lymphomas in general.

→ We strongly disagree. Different from very old references quoted here by the referee, we and others found that E μ -myc lymphomas originate from both immature and mature B-cells (Schmitt-CA et al., Genes Dev 1999) that accomplished clonal V(D)J recombination and progress to a mature, CD43-negative but frequently surface-IgD-positive B-cell state (Reimann-M et al., Blood 2007, and unpublished data), with activation-induced cytidine deaminase – conferring somatic hypermutations – being required in the E μ -myc lymphoma model for the development of mature B-cell lymphomas (Kotani-A et al., PNAS 2007), all indicative of their germinal center relationship.

There is ample evidence in the more recent literature that underscores both the close recapitulation of human DLBCL by the E μ -myc transgenic mouse lymphoma model and the value of this model to uncover novel molecular mechanisms relevant for DLBCL biology and its clinical behavior. Just to list a few examples:

1. “...E μ -myc tumor... ...show similarity with human diffuse large B-cell lymphoma in the pattern of gene expression, as well as oncogenic pathway activation... ...signatures of oncogenic pathway activity provide further dissection of the spectrum of diffuse large B-cell lymphoma, identifying a subset of patients who have very poor prognosis and could benefit from more aggressive or novel therapeutic strategies...” (Mori-S et al., Cancer Res 2008)
2. “...Figure 5. Human diffuse large B-cell lymphomas (DLBCL) display features consistent with the [E μ -myc-uncovered] model of non-cell-autonomous TGF- β -mediated cellular senescence...” (Reimann-M et al., Cancer Cell 2010)
3. “Further characterization and genetic engineering of primary [E μ -myc] mouse lymphomas according to distinct NF- κ B-related oncogenic networks reminiscent of diffuse large B-cell lymphoma (DLBCL) subtypes guided us to identify Bcl2-overexpressing germinal center B-cell-like (GCB) DLBCL as a clinically relevant subgroup with significantly superior outcome when NF- κ B is hyperactive...” (Jing-H et al., Genes Dev 2011)
4. “...Fig. 2C: box and whisker plot of GCB-ABC signature scores in cluster 1 and cluster 2 E μ -myc lymphomas”... ...DLBCL is most similar to cluster 2 E μ -myc lymphomas (Fig. 3). Together with the analysis of genomic data with regard to GCB versus ABC distinction, these results define... ...cluster 2 E μ -myc lymphoma as a representation of the ABC subtype of human DLBCL... ...we found significant distinctions in cellular pathway activity... ...whereas TGF β , STAT3, TNF α , EGFR, and IFN pathways are significantly upregulated in DLBCL... ...a similar pattern of pathway activity is seen... ...cluster 2 E μ -myc lymphoma samples as is seen... ...DLBCL (Fig. 6B). To quantitate the similarity, we calculated binary logistic regression coefficients of the genomic signatures with respect to the human and E μ -myc lymphoma, and found a significant correlation between the coefficients for the lymphomas ($r = 0.961$, Pearson correlation test, Fig. 6C)...” (Rempel-RE et al., Mol Cancer Ther 2014)
5. “...Fig. 3f, Nuclear β -catenin expression by immunostaining of lymph nodes from [E μ -myc] control;Bcl2 lymphoma-bearing mice... ...and human DLBCL biopsies from the same individual patients at diagnosis and at relapse after first-line induction chemotherapy...” (Milanovic-M et al., Nature 2018)
6. “...we combine the *Blim*^{m3/m3} allele with an inactivating *PB* transposon system in [E μ -myc] mice to achieve genome-wide tumor suppressor gene screening in B-cell lymphoma. We identify known and novel DLBCL genes, validate selected candidate genes through a CRISPR/Cas9-

based functional approach and show the clinical relevance of our findings using large human DLBCL patient cohorts...” (Weber-J et al., Nature Comm 2019)

7. “Analyses of... brains infiltrated with Eμ-myc cells *via* immunohistochemistry and qPCR revealed an expression profile consistent with DLBCL, including Myc, IRF4, Bcl2, and immunoglobulin M (Figures S2B–S2F)... ...Brains from mice injected with Eμ-myc... ...lymphoma cells exhibited an expression profile consistent with DLBCL (Figures 3F and S3D–S3G)... ...Fig. 3F: mRNA or protein (Ki67) expression profile of Eμ-myc... ...brain lesions compared with human DLBCL and Burkitt lymphoma (BL), another Myc-driven malignancy [showing high similarity between Eμ-myc and DLBCL samples, but not between Eμ-myc and BL samples]...” (O’Connor-T et al., Cancer Cell 2019)

Moreover, while diffuse large B-cell lymphomagenesis is undoubtedly intimately linked to selective mechanisms relevant for normal B-cell ontogenesis in the germinal center (GC) reaction, no mouse model has been generated so far in which putative GC B-cells or activated B-cells (ABC) are faithfully converted into a DLBCL-reminiscent malignancy – as postulated by the transcriptome signature-based discrimination of an GCB- vs. ABC-subtype-distinct cell-of-origin (COO). One of the underlying reasons might be the possibility that precursor lesions of GC lymphomas actually acquire their initiating hits prior to their GC passage in secondary lymphoid organs, *e.g.* in the bone marrow, and perhaps sometimes as part of the so called “clonal hematopoiesis of indeterminate potential (CHIP)” (Genovese-G et al., NEJM 2014; Jaiswal-S et al., NEJM 2014) with involvement of genes associated with lymphoid tumorigenesis (*e.g.* TET2, SF3B1, TP53 or IDH2), or even at the level of CD34+ hematopoietic stem cells (Damm-F et al., Cancer Discovery 2014).

Nevertheless, we profoundly addressed the question of the representativeness of Eμ-myc lymphomas for human DLBCL at the beginning of our investigation, and provided very strong evidence of their proximity to human DLBCL but much less to human Burkitt’s lymphoma in Fig. 1c as a transcriptome-based principal component analysis utilizing large sample numbers. Furthermore, the excellent recapitulation of the DLBCL-targeting Shipp lab-based “comprehensive consensus cluster (CCC)” DLBCL classification by Eμ-myc lymphomas (Fig. 1e) is another, COO signature-independent piece of evidence that underscores the biological proximity between the DLBCL molecular architecture and the Eμ-myc model.

In essence, the Eμ-myc model reflects a close approximation to typical histological, biochemical, genetic and clinical features of human DLBCL, especially features based on maturity, somatic hypermutation, Myc activation, GCB/ABC COO subtype designation, the more functional CCC classification, and NF-κB pathway activation, just to name a few, and has been instrumental in the past to primarily uncover genetic defects and fundamental biological principles in this mouse model that were subsequently confirmed by validation analyses in human DLBCL specimens.

As the EmuMYC tumors are not GC derived, it is unclear how the authors could classify them in ABC-

DLBCL and GCB-DLBCL (the data in Figure S1C are indeed weak). Mouse GC B cells and activated B cells should be used as a training set to define the linear predictor score, as done for the human tumors in Alizadeh et al 2001. Differences in NF- κ B transcriptional signatures may be reminiscent of certain aspects of ABC-DLBCL but they are not sufficient to classify a tumor into the ABC-subtype of DLBCL given that this pathway is engaged in a multitude of cancers.

→ Since the assumption of the referee – $E\mu$ -*myc* lymphomas not being GC-derived – does not appear, as discussed above, to be generally correct, our approach is not flawed but adequately applies a murinized version of the original Staudt lab “Wright” classifier, well-established as a linear predictor score to assign human DLBCL as either GCB-, unclassifiable or ABC-subtype lymphomas, to identify $E\mu$ -*myc* lymphomas with higher expression of GCB-reminiscent and $E\mu$ -*myc* lymphomas with higher expression of ABC-typical transcripts – which apparently worked well (by no means “indeed weak”) as shown in Supplementary Fig. 1c, and further functionally validated with results in closest proximity to human DLBCL (Fig. 1c, Fig. 5f). The referee is also mistaken if assuming we would have used an NF- κ B signature to assign here $E\mu$ -*myc* lymphomas as being GCB- or ABC-like – this was done in a former paper (Jing-H et al., *Genes Dev* 2011), but not here, where we strictly adhered to the Staudt lab-inaugurated classifier.

The parallel made between the pattern of response/relapse to cyclophosphamide in the mouse model and the behavior of DLBCL patients is also misleading and cannot be taken as an evidence for the EmMYC tumors to “recapitulate treatment outcome of DLBCL”. Cyclophosphamide is used in several regimens in addition to R-CHOP, and the high initial chemosensitivity with relapse in a subset of mice and progressively reduced response to secondary treatments is a feature of many experimental systems.

→ The superior outcome of mice bearing human-to-mouse classifier-based GCB-assigned $E\mu$ -*myc* lymphomas and the inferior outcome of those harboring ABC-assigned $E\mu$ -*myc* lymphomas to a central component of the CHOP regimen used to treat DLBCL patients with more favorable outcome of GCB- as compared to ABC-subtype patients is by no means “misleading”, but must be taken as strong evidence for the proximity between the lymphoma biology of both species. That Cyclophosphamide may produce high responder rates with frequent failures over time in other experimental systems may indicate its entity-overarching but still biology/genetics-dependent efficacy, but is not relevant here. The central point here is that the human-to-mouse informed GCB-subtype mouse lymphomas do much better than their ABC-subtype counterparts, based on the same genetic stratifier principle that has changed the clinical perception of DLBCL patients as GCB vs. Non-GCB/ABC since the beginning of the millennium. This is certainly not a “general” or highly anticipatable effect of Cyclophosphamide – it is an experimental demonstration of the strong biological proximity between GCB- and Non-GCB/ABC-like mouse lymphoma subtypes and their human COO counterparts.

Statements such as “these lymphomas recapitulate clinical, histopathological and genetic features of human lymphoma” or are “faithful models of DLBCL” or have “genetic and clinical proximity to DLBCL” are incorrect and confusing.

→ As explicated above, we must disagree, since these statements are simply correct. None of these statements is saying that Eμ-*myc* lymphomas “fully recapitulate” human DLBCL or represent a “perfect model” of human DLBCL. Their proximity to the human condition is indeed striking and the basis of this and many other colleagues’ highly meaningful cross-species research. And: Eμ-*myc* lymphomas, even if missing certain aspects of human DLBCL biology, have been provenly instrumental in elucidating novel mechanisms and principles, whose relevance was subsequently molecularly and clinically validated in the human condition. Hence, even a “non-believer” might acknowledge the power of functional genetics in the Eμ-*myc* lymphoma system to generate hypotheses that – of course – require confirmatory research in DLBCL (as we do here) before any conclusion should be drawn. Nevertheless, we take the concerns of the referee serious, and have carefully re-checked the manuscript for any potential overstatement in this regard, and toned down specific statements on the suitability of the Eμ-*myc* model for human DLBCL in the revised version wherever it deemed to be appropriate.

The manuscript is difficult to read; limited use of adverbs and shorter sentences are recommended.

→ We are grateful for this remark of the referee and revised the manuscript accordingly, *i.e.* keeping sentences shorter, and reduced the use of adverbs, as suggested.

REVIEWERS' COMMENTS:

Reviewer #1 (Remarks to the Author):

The revised version of this manuscript has addressed the majority of my concerns from the first submission, and the answers to my questions are insightful and generally convincing.

Having said that, I had liked to see more data on the crosstalk between senescence and apoptosis (Figure 3) and further quantifications, while important questions for future works still remain open. As an example, apart from the possible scenarios stated by the authors (i.e. stochastic process, signaling network-governed or competition), it would be interesting to know whether control mice between 10d CTX and 30d CTX benefit or not from a process of immune surveillance, or if senescence and apoptosis coexist. Although I agree with the authors that the point of competition between senescence and apoptosis is out of the scope of the manuscript, I think the results on the competing ability of the cells to enter senescence upon TIS (e.g. 10d CTX) would benefit at least from an additional dissection of markers of senescence and quantifications.

Overall, and despite some limitations, I think this article is an elegant and well-executed work that provides interesting predictive tools including "SUVARness", a senescence-indicating gene signature, as well as high-levels of H3K9m3, which have the potential to be exploited for favorable prediction of DLBCL patient outcome in the context of TIS. I have no further comments or suggestions and congratulate the authors, as the manuscript has been significantly improved.

Reviewer #4 (Remarks to the Author):

This revised version of the manuscript by Schleich and colleagues, including a point-by-point letter of response to reviewers, reports that E μ -myc transgenic lymphomas recapitulate transcriptional signatures, including COO and CCC classifications, of human DLBCLs. E μ -myc lymphomas treated in vivo with CHOP chemotherapy show survival responses that may resemble those of DLBCL patients treated with CHOP/R-CHOP. Further studies using these models with manipulation of the methyltransferase Suv39h1 and H3K9me3-active demethylase in loss- and gain-of-function assays identify a mouse-derived senescence-indicating gene signature and high expression of H3K9me3 that predict a favorable outcome of patients with DLBCL.

I find the manuscript of interest, as it provides data on using a defined mouse model that recapitulate features of human tumors, leading to the identification of cellular senescence as a biomarker that may predict outcome of human DLBCL. The experiments are well designed and performed, the manuscript is well written, and the conclusions of the study are sustained by the data presented.

My major concern is, like reviewer #2, up to what point E μ -myc transgenic lymphomas parallel the genetics and the biology of human DLBCL. Theoretically, they do not, as there are clear differences in the genetic and cellular origins, and pathology, of these mouse lymphomas vs. human DLBCLs. E μ -myc lymphomas are originated by one single genetic defect that is activated at early B cells and leads to the development of lymphoblastic B cell lymphomas/leukemias, while the genetically heterogeneous human DLBCLs have in common a more mature cellular origin (germinal or post-germinal center B cells), but differ in the nature of the genetic lesions and signaling pathways that collectively become deregulated to drive the full malignant phenotype (as shown in the original COO/CCC papers by Staudt and Shipp, and by the more recent "cluster" and "BN2/MCD/EZB/N1" classifications).

In this manuscript, however, the transcriptional signatures of E μ -myc lymphomas clearly overlap with those of human DLBCL, as shown in Fig1c and 1e, and moreover, these expression profiles disclose survival curves that appear similar to those of DLBCL patients (Fig.1b and 1d). Likewise, the transcriptional signatures of responders vs non-responder mouse lymphomas (Fig.2) seem also to separate patients with DLBCL into those with favorable and more unfavorable OS/EFS upon R-CHOP therapy. Therefore, while these data in mice seem to be in contrast of what we have learnt

about the biology and genetics of DLBCL during the last 2 decades, the manuscript provide experimental evidence supporting that E μ -myc lymphomas develop tumors that, at least molecularly, recapitulate human DLBCL features. In other words, although theoretically E μ -myc lymphomas and human DLBCLs represent totally different tumors, the experimental data presented here seem to validate the mice as valid models to interrogate human biology. While I think that the E μ -myc model is not perfect at all, its experimental use here leads to the major conclusion of the manuscript, which is that H3K9me3-governed senescence may predict lymphoma patient outcome.

There are more representative and much better transgenic mouse models developing human-like DLBCL, including both ABC and GCB subtypes, which have been generated using a rationally-designed genetic approach, and have been thereby accepted by the scientific community as valid experimental models to study human biology. For instance, ABC-DLBCLs arise in mice with constitutive activation of NF- κ B signaling and Blimp1 deletion (Calado et al, Cancer Cell 2010; Pascual et al, Blood 2019), genetic changes that are consistently found in ABC DLBCL patients. In addition, lymphomas with features of human GCB DLBCL arise in mice with Bcl2 expression and Crebbp genetic inactivation (Zang et al, Cancer Discov 2017; Jiang et al, Cancer Discov 2017; Garcia-Ramirez et al, Blood 2017), two common genetic features of human GCB DLBCL. While I think that the implication of cellular senescence could be interrogated in these models, the results shown here, indicating that single Myc expression may recapitulate (at least in part) the complexity and heterogeneity of human DLBCL, are enigmatic.

Said that, I think that the authors should tone down their comments and conclusions in the abstract and throughout the text in regard of the similarities and overlapping features of E μ -myc lymphomas and human DLBCLs. For example, I would suggest changing the sentence "We present here how primary E μ -myc transgenic lymphomas faithfully recapitulate molecular features and clinical courses of patients diagnosed with diffuse large B-cell lymphoma (DLBCL)" by "We present here how primary E μ -myc transgenic lymphomas recapitulate molecular signatures of patients diagnosed with diffuse large B-cell lymphoma (DLBCL), which predict responses to chemotherapy". Additionally, I agree with reviewer#2 in his/her comments on the parallelism made between the pattern of response/relapse to cyclophosphamide in the mouse model and the clinical behavior of DLBCL patients upon therapy (Figure 2). I rather think that the progressive loss of response of E μ -myc lymphomas to chemotherapy is simply a common feature of all cancers (not particularly of DLBCL). This is probably even more evident with current therapy of DLBCL (rituximab+CHOP, R-CHOP), as CHOP is not used alone anymore. I think these Fig.2 comments should not be included in the manuscript. I would also suggest including in the discussion section the limitations and controversies indicated by the reviewer #2 and by myself here on the E μ -myc mice as a model to characterize human DLBCL pathology.

Point-by-point response (highlighted in blue) to the reviewers' comments

Reviewer #1 (Remarks to the Author); expert in senescence and cancer, mouse models:

The revised version of this manuscript has addressed the majority of my concerns from the first submission, and the answers to my questions are insightful and generally convincing.

Having said that, I had liked to see more data on the crosstalk between senescence and apoptosis (Figure 3) and further quantifications, while important questions for future works still remain open. As an example, an apart from the possible scenarios stated by the authors (i.e. stochastic process, signaling network-governed or competition), it would be interesting to know whether control mice between 10d CTX and 30d CTX benefit or not from a process of immune surveillance, or if senescence and apoptosis coexist. Although I agree with the authors that the point of competition between senescence and apoptosis is out of the scope of the manuscript, I think the results on the competing ability of the cells to enter senescence upon TIS (e.g. 10d CTX) would benefit at least from an additional dissection of markers of senescence and quantifications.

→ We commented on this point in the previous rebuttal. Whether fate decisions between apoptosis vs. senescence are merely stochastic, or signaling network-governed, or the product of actual “cellular competition” is an intriguing question raised here by the referee that we are actively pursuing, with required experiments and techniques (e.g. single-cell RNA-seq, fate tracking, 3D-analysis of receptor/ligand interactions etc.) going well-beyond the scope of the current manuscript. The referee now mentioned with “immune surveillance” an additional layer to this complex question, which we couldn’t agree more on, a very hot topic we also currently actively work on, but another theme for which satisfying answers clearly go beyond the scope of the current manuscript. For exactly these reasons, we kept Fig. 3 scientifically focused on cellular senescence, specifically, whether a detectable residual tumor load in the absence of a Bcl2-mediated apoptotic block is controlled by senescence, and whether such arrest program would contribute to long-term outcome or not. Data on virtually indistinguishable *in vitro*-kinetics of drug-induced cell death in non-bcl2-protected lymphomas with a senescence-capable (“control”) vs. a senescence-incapable (“Suv39h1-deficient”) genotype are provided in Supplementary Fig. 3c – if viewed in light of the quantitative differences seen at day 10 *in vivo* (Fig. 3), it becomes clear that no “snap-shot” analyses of a given apoptosis or senescence rate during these ten days could accurately quantify the subtle imbalances that, as a net effect, lead to the quite profound differences in residual tumor burden and presence or absence of senescent cells therein at day 10 – for which we provide robust quantification.

Overall, and despite some limitations, I think this article is an elegant and well-executed work that provides interesting predictive tools including "SUVARness", a senescence-indicating gene signature, as well as high-levels of H3K9m3, which have the potential to be exploited for favorable prediction of DLBCL patient outcome in the context of TIS. I have no further comments or suggestions and congratulate the authors, as the manuscript has been significantly improved.

→ We are grateful to the referee’s thoughtful comments and repeated willingness to comment so carefully on our work. We are delighted that the revision has led to a significantly improved version of the manuscript that did not prompt further suggestions by this referee.

New Reviewer #4 (Remarks to the Author):

This revised version of the manuscript by Schleich and colleagues, including a point-by-point letter of response to reviewers, reports that E μ -myc transgenic lymphomas recapitulate transcriptional signatures, including COO and CCC classifications, of human DLBCLs. E μ -myc lymphomas treated in vivo with CHOP chemotherapy show survival responses that may resemble those of DLBC patients treated with CHOP/R-CHOP. Further studies using these models with manipulation of the methyltransferase Suv39h1 and H3K9me3-active demethylase in loss- and gain-of-function assays identify a mouse-derived senescence-indicating gene signature and high expression of H3K9me3 that predict a favorable outcome of patients with DLBCL.

I find the manuscript of interest, as it provides data on using a defined mouse model that recapitulate features of human tumors, leading to the identification of cellular senescence as a biomarker that may predict outcome of human DLBCL. The experiments are well designed and performed, the manuscript is well written, and the conclusions of the study are sustained by the data presented.

My major concern is, like reviewer #2, up to what point E μ -myc transgenic lymphomas parallel the genetics and the biology of human DLBCL. Theoretically, they do not, as there are clear differences in the genetic and cellular origins, and pathology, of these mouse lymphomas vs. human DLBCLs. E μ -myc lymphomas are originated by one single genetic defect that is activated at early B cells and leads to the development of lymphoblastic B cell lymphomas/leukemias, while the genetically heterogeneous human DLBCLs have in common a more mature cellular origin (germinal or post-germinal center B cells), but differ in the nature of the genetic lesions and signaling pathways that collectively become deregulated to drive the full malignant phenotype (as shown in the original COO/CCC papers by Staudt and Shipp, and by the more recent “cluster” and “BN2/MCD/EZB/N1” classifications).

→ It's probably only fair to clarify that E μ -myc lymphomas have one driving transgenic oncogene but are, of course, much more complex and diverse in their genetic make-up (means: definitely require cooperating secondary genetic hits), as reported by many colleagues and reflected by their long and highly variable latency to tumor onset. It is probably exactly this broad spectrum of early to rather mature B-cell lymphomas related to their diverse secondary hits that accounts for a heterogeneity in the E μ -myc model which seems to resemble the heterogeneous DLBCL genetics.

In this manuscript, however, the transcriptional signatures of E μ -myc lymphomas clearly overlap with those of human DLBCL, as shown in Fig1c and 1e, and moreover, these expression profiles disclose survival curves that appear similar to those of DLBCL patients (Fig.1b and 1d). Likewise, the transcriptional signatures of responders vs non-responder mouse lymphomas (Fig.2) seem also to separate patients with DLBCL into those with favorable and more unfavorable OS/EFS upon R-CHOP therapy. Therefore, while these data in mice seem to be in contrast of what we have learnt about the biology and genetics of DLBCL during the last 2 decades, the manuscript provide experimental evidence supporting that E μ -myc lymphomas develop tumors that, at least molecularly, recapitulate human DLBCL features. In other words, although theoretically E μ -myc lymphomas and human DLBCLs represent totally different tumors, the experimental data presented here seem to validate the mice as valid models to interrogate human biology. While I think that the E μ -myc model is not perfect at all, its experimental use here leads to the major conclusion of the manuscript, which is that H3K9me3-governed senescence may predict lymphoma patient outcome.

→ We are grateful for this data-open view of the referee. As our data indicate throughout the manuscript (just to mention Fig. 1c and 1e), it becomes apparent that E μ -myc lymphomas and

human DLBCL, although viewed for understandable reasons as “theoretically totally different tumors”, factually possess critical similarities.

There are more representative and much better transgenic mouse models developing human-like DLBCL, including both ABC and GCB subtypes, which have been generated using a rationally-designed genetic approach, and have been thereby accepted by the scientific community as valid experimental models to study human biology. For instance, ABC-DLBCLs arise in mice with constitutive activation of NF- κ B signaling and Blimp1 deletion (Calado et al, Cancer Cell 2010; Pascual et al, Blood 2019), genetic changes that are consistently found in ABC DLBCL patients. In addition, lymphomas with features of human GCB DLBCL arise in mice with Bcl2 expression and Crebbp genetic inactivation (Zang et al, Cancer Discov 2017; Jiang et al, Cancer Discov 2017; Garcia-Ramirez et al, Blood 2017), two common genetic features of human GCB DLBCL. While I think that the implication of cellular senescence could be interrogated in these models, the results shown here, indicating that single Myc expression may recapitulate (at least in part) the complexity and heterogeneity of human DLBCL, are enigmatic.

→ Accepted – and the landscape of findings indeed remains somewhat puzzling at this moment of intense scientific investigations. While there are by now additional and perhaps “better” DLBCL-approaching mouse models developed, it’s again only fair to say that any specific gene sequence or combination chosen as ABC- or GCB-subtype DLBCL approximation intrinsically cannot represent the actual heterogeneity the human disease presents with. Moreover – but clearly going beyond the scope of this manuscript and its re-rebuttal – the ABC/GCB distinction is a linear regression/linear predictor score-based bioinformatics dissection of transcriptome data that has limited direct biological, or, more specifically, B-cell ontogeny-anchored reflection. What truly matters and what our work provides major conceptual advance about is the contribution of biological programs to tumor fate and outcome in response to therapeutic challenges. The genetic determinants identified in the mouse model system chosen to functionally interrogate senescence establish some scientific merit on their own – but their positive verification in human DLBCL datasets, disconnected from a scientific debate about the model quality *per se*, is a key discovery and a stimulus towards future clinical research and therapeutic intervention in the DLBCL-caring community. Why lymphomas with “single Myc expression” (*i.e.* a single oncogenic lesion to begin with) may recapitulate complexity and heterogeneity of human DLBCL is, in our view, not at all “enigmatic” but explained by the high genetic variability of $E\mu$ -myc lymphomas and their broad representation across the B-cell ontogenicity cascade from immature to quite mature malignancies.

Said that, I think that the authors should tone down their comments and conclusions in the abstract and throughout the text in regard of the similarities and overlapping features of $E\mu$ -myc lymphomas and human DLBCLs. For example, I would suggest changing the sentence “We present here how primary $E\mu$ -myc transgenic lymphomas faithfully recapitulate molecular features and clinical courses of patients diagnosed with diffuse large B-cell lymphoma (DLBCL)” by “We present here how primary $E\mu$ -myc transgenic lymphomas recapitulate molecular signatures of patients diagnosed with diffuse large B-cell lymphoma (DLBCL), which predict responses to chemotherapy”. Additionally, I agree with reviewer#2 in his/her comments on the parallelism made between the pattern of response/relapse to cyclophosphamide in the mouse model and the clinical behavior of DLBCL patients upon therapy (Figure 2). I rather think that the progressive loss of response of $E\mu$ -myc lymphomas to chemotherapy is simply a common feature of all cancers (not particularly of DLBCL). This is probably even more evident with current therapy of DLBCL (rituximab+CHOP, R-CHOP), as CHOP is not used alone anymore. I think these Fig.2 comments should not be included in the manuscript. I would also suggest including in the discussion section the limitations and controversies indicated by the reviewer #2 and by myself here on the $E\mu$ -myc mice as a model to characterize human DLBCL pathology.

→ We have no objections to follow the suggestions of this referee, and, hence, toned down statements that might have been misunderstood as a euphoric appraisal of the E μ -myc model to closely recapitulate molecular features of human DLBCL. We also carefully revised our wording regarding the course of Cyclophosphamide-failing E μ -myc lymphomas and induction-failing DLBCL patients, although, in our opinion, not many mouse models present with such patient-reminiscent behavior when subjected to repeated chemotherapy exposure, and even exhibit crucial distinct molecular commonalities in the relapse situation (Wnt signaling, for instance), as we recently reported (Milanovic-M et al., Nature 2018). And, lastly, we also enhanced the discussion in the re-revised version of the manuscript by a more balanced view on the limitations and controversies arising from model-centered criticism of our work.